# Tripartite motif containing 26 prevents steatohepatitis progression by suppressing C/EBPδ signalling activation

Minxuan Xu [1,2,7] ✉, Jun Tan [1,7] ✉, Xin Liu[3,7], Li Han[3,7], Chenxu Ge[1,2,7], Yujie Zhang[1], Fufang Luo[1], Zhongqin Wang[1], Xiaoqin Xue[1], Liangyin Xiong[1], Xin Wang[1], Qinqin Zhang[1], Xiaoxin Wang[1], Qin Tian[1], Shuguang Zhang[3], Qingkun Meng[4], Xianling Dai[1,2], Qin Kuang[1,2], Qiang Li[1], Deshuai Lou[1], Linfeng Hu[1,2], Xi Liu[1], Gang Kuang[1], Jing Luo[1], Chunxiao Chang[4], Bochu Wang[2], Jie Chai[4], Shengbin Shi[5] ✉ & Lianyi Han[6] ✉

Currently potential preclinical drugs for the treatment of nonalcoholic steatohepatitis (NASH) and NASH-related pathopoiesis have failed to achieve expected therapeutic efficacy due to the complexity of the pathogenic mechanisms. Here we show Tripartite motif containing 26 (TRIM26) as a critical endogenous suppressor of CCAAT/enhancer binding protein delta (C/ EBPδ), and we also confirm that TRIM26 is an C/EBPδ-interacting partner protein that catalyses the ubiquitination degradation of C/EBPδ in hepatocytes. Hepatocyte-specific loss of Trim26 disrupts liver metabolic homeostasis, followed by glucose metabolic disorder, lipid accumulation, increased hepatic inflammation, and fibrosis, and dramatically facilitates NASH-related phenotype progression. Inversely, transgenic Trim26 overexpression attenuates the NASH-associated phenotype in a rodent or rabbit model. We provide mechanistic evidence that, in response to metabolic insults, TRIM26 directly interacts with C/EBPδ and promotes its ubiquitin proteasome degradation. Taken together, our present findings identify TRIM26 as a key suppressor over the course of NASH development.

The ascending prevalence of obesity and its associated complications has been treated as a global pandemic in many countries. As the obese population continues to rise around the world, the morbidity and mortality of obesity-associated complications, e.g., obstractive sleep apnea (OSA), type 2 diabetes (T2D), hyperuricemia, nonalcoholic fatty liver disease (NAFLD), stroke, tumorigenesis, irregular menstrual cycles (IMC), etc., have dramatically increased in parallel, and NAFLD affects around 20–30% of the adult population and 70% of the global hypertension, diabetes, and obesity population[1–5]. The spectrum of NAFLD pathology encompasses a spectrum of manifestations, ranging

[1]Chongqing Key Laboratory of Medicinal Resources in the Three Gorges Reservoir Region, School of Biological and Chemical Engineering, Chongqing University of Education, 400067 Chongqing, P. R. China. [2]Key Laboratory of Biorheological Science and Technology (Chongqing University), Ministry of Education, College of Bioengineering, Chongqing University, 400030 Chongqing, P. R. China. [3]Department of Gastrointestinal Surgery, Shandong Cancer Hospital and Institute, Shandong First Medical University & Shandong Academy of Medical Science, 250117 Jinan, P. R. China. [4]Geriatrics Department, The Second Affiliated Hospital of Shandong University of Traditional Chinese Medicine, 250117 Jinan, P. R. China. [5]New Drug Technology R&D Center, Nanjing Biomed Sciences Inc., 210003 Nanjing, P. R. China. [6]Greater Bay Area Institute of Precision Medicine (Guangzhou), School of Life Sciences, Fudan University, 315211 Shanghai, China. [7]These authors contributed equally: Minxuan Xu, Jun Tan, Xin Liu, Li Han, Chenxu Ge. ✉e-mail: minxuanxu@foxmail.com; tanjun@cque.edu.cn; shisb_njbms@yeah.net; hanlianyi@fudan.edu.cn

from benign hepatosteatosis to more severe forms characterised by hepatocellular injury and necroinflammatory changes, as evidenced by the presence of nonalcoholic steatohepatitis (NASH). This suggests that individuals with NAFLD are at an increased risk of developing liver fibrosis and hepatocellular carcinoma (HCC), and are more prone to experiencing these pathological progressions[6]. Nevertheless, there remains a lack of comprehensive understanding regarding the mechanisms behind the pathophysiology of non-alcoholic steatohepatitis (NASH). Currently, there is a lack of globally licenced and substantiated therapies for nonalcoholic steatohepatitis (NASH), and research endeavours aimed at exploring the complexities linked to NASH have not fully fulfilled anticipated outcomes. In recent years, a series of candidate preclinical medicines for non-alcoholic steatohepatitis (NASH), such as emricasan (VAY785), selonsertib (GS-4997), and pegbelfermin (BMS-986036), have encountered setbacks in clinical trials due to the failure to meet desired clinical objectives[7–9]. The lack of success observed in these potentially effective preclinical medications offers additional substantiation for the notion that the development and advancement of non-alcoholic steatohepatitis (NASH) are characterised by intricate molecular mechanisms and may be traced through numerous physiological and metabolic pathways. Hence, it is imperative to conduct additional extensive research in order to precisely ascertain the principal molecular regulators responsible for the progression of non-alcoholic steatohepatitis (NASH), with the ultimate goal of formulating enhanced therapy protocols.

E3 ubiquitin ligase tripartite motif-containing protein 26 (TRIM26, also known as RNF95 or ZNF173) has been determined as a multifunctional regulator in innate immune response and chronic metabolic disease development by regulating the targeted substrate ubiquitination modification[10,11]. Previous studies have confirmed that, as a remarkably negative regulator, TRIM26 retards proliferation, migration, metastasis, and glycolysis in papillary thyroid carcinoma (PTC) and HCC via suppression of PI3K/AKT signalling or regulation of ZEB1 ubiquitination[12,13]. Furthermore, TRIM26 protects against hepatic stellate cell activation and thereby alleviates the progression of hepatic fibrosis by mediating SLC7A11 ubiquitination and ferroptosis[14]. Notably, in the innate immune response, forced activation of TRIM26 is capable of inhibiting interferon-β production and the corresponding antiviral response by promoting polyubiquitination and degradation of nuclear IRF3[11]. However, another report determined that TRIM26 effectively facilitated the innate immune response against RNA virus infection by recruiting NEMO to promote its interaction with MAVS and TBK1[15]. Lastly, TRIM26 also positively regulates proinflammatory responses by catalysing K11-linked ubiquitination of TAB1 and promoting NF-κB and MAPK signalling activation in a colitis rodent model[10]. These distinct findings reported that TRIM26 actually performed different molecular biological functions in unusual physiological and pathological processes by sharing different molecular mechanisms. Additionally, the function of TRIM26, especially in NASH pathogenesis, remains elusive, so these facts compel us to investigate the potential molecular mechanism of TRIM26.

In this study, our objectives are to (i) examine the relationship between TRIM26 expression in livers and the severity of non-alcoholic steatohepatitis (NASH) in patients, (ii) investigate the impact of altering Trim26 activity on NASH-related processes such as lipid deposition, inflammation, and fibrosis using various dietary rodent models of NASH combined with loss-of-function and gain-of-function transgenic expression mice, and (iii) elucidate the downstream signalling pathways regulated by Trim26 in the suppression of NASH pathogenesis through a series of in vivo experiments involving adeno-associated virus (AAV)-mediated knockdown and in vitro experiments using adenovirus methods. Therefore, we have established that there is a notable reduction in TRIM26 levels in liver tissues, which exhibits a negative association with the expression of markers indicative of non-alcoholic steatohepatitis (NASH) in both human subjects and a mouse model of NASH. In a set of mice models exhibiting non-alcoholic steatohepatitis (NASH), the targeted removal of TRIM26 specifically in hepatocytes resulted in a substantial increase in hepatic steatosis, liver inflammation, and hepatofibrosis. Conversely, the overexpression of TRIM26 in hepatocytes greatly mitigated the development of NASH diet-induced steatohepatitis. In our study, we investigated the mechanisms by which TRIM26 responds to metabolic insults. We found that TRIM26 interacts directly with CCAAT/enhancer binding protein delta (C/EBPδ, CEBPD) and hinders its activation by facilitating its degradation through the ubiquitin proteasome pathway. As a result, this interaction prevents the activation of C/EBPδ-HIF1A signalling and the subsequent activation of downstream pathways. The aforementioned findings indicate a significant molecular mechanism that governs the activation of CEBPD throughout the advancement of non-alcoholic steatohepatitis (NASH). Furthermore, these findings reveal a potential avenue for targeted therapy of NASH and its related consequences, offering a viable therapeutic strategy.

## Results

### Hepatocyte TRIM26 expression is inversely correlated with NASH severity in a rodent model and human subjects

As a means to examine the role of ubiquitin ligases in the setting of non-alcoholic steatohepatitis (NASH), we conducted an investigation to assess their expression levels in liver tissue obtained from mice that were induced to develop NASH through a high-energy diet, as well as from human patients exhibiting the clinical phenotype of NASH. In accordance with the findings presented in Fig. 1a, b, our study observed that the administration of a 0.5 mM palmitic acid+1.0 mM oleic acid (PAOA) mixture, along with 5 mM fructose (Fru), to human L02 cells or primary hepatocytes, 16-week high-fat high-cholesterol (HFHC) or Western-type diet+fructose drinking water (WTDF)-fed WT mice, and non-alcoholic steatohepatitis (NASH) patient samples resulted in the identification of four distinct ubiquitin-associated ligases. These ligases, namely TRIM26, RNF5, TRIM31, and TRIM8, were found to be prominently expressed in the overlapping experimental groups. Then, the differential expression genes (DEGs) in the two species of PAOA-induced hepatocytes over time were analysed. As expected, 4 E3 ubiquitin ligase indicators, i.e., *TRIM26*, *RNF5*, *TRIM31*, and *TRIM8*, met the selection criteria for intersection screening, among which *TRIM26* displayed the most significant downregulation in the whole analysis (Fig. 1c). Additionally, it was observed that the expression patterns of TRIM26 protein and its corresponding mRNA were progressively decreased in the livers from 0 to 16 weeks following HFHC treatment, as indicated by both dynamic expression analysis and examination of human liver samples ($P < 0.01$ according to one-way ANOVA; Fig. 1d, Supplementary Fig. 1a–d). In addition, we conducted an investigation into the suppression of TRIM26 expression by PAOA-mediated lipidic toxicity. This investigation involved integrating the prediction of transcription factors (TFs) that bind to the TRIM26 promoter, utilising human and mouse GSE databases and conjoint analysis (Supplementary Fig. 1e). A critical transcription factor, HNF4A, was highlighted by a united analysis among the Top 10 predicted TFs of each database. To further examine HNF4A, which is responsible for TRIM26 expression in hepatocytes, the hepatocytes were transfected with luciferase reporter plasmids containing human TRIM26 promoter truncations or its mutants, accompanied by the corresponding pRL-TK control vector, to examine the relative luciferase activity. It was observed that the activation of the TRIM26 promoter by HNF4A remained intact even after excision of the 5′-end up to −925 bp. However, the luciferase activity induced by HNF4A was entirely inhibited when the nucleotide excision was extended to −205 bp ($P < 0.0001$ by two-tailed $t$ test; Supplementary Fig. 1f). The results of this study have established that the sequence spanning from −674 bp to −649 bp in the TRIM26 promoter region is the critical region responsible for the transcriptional induction triggered by HNF4A. The

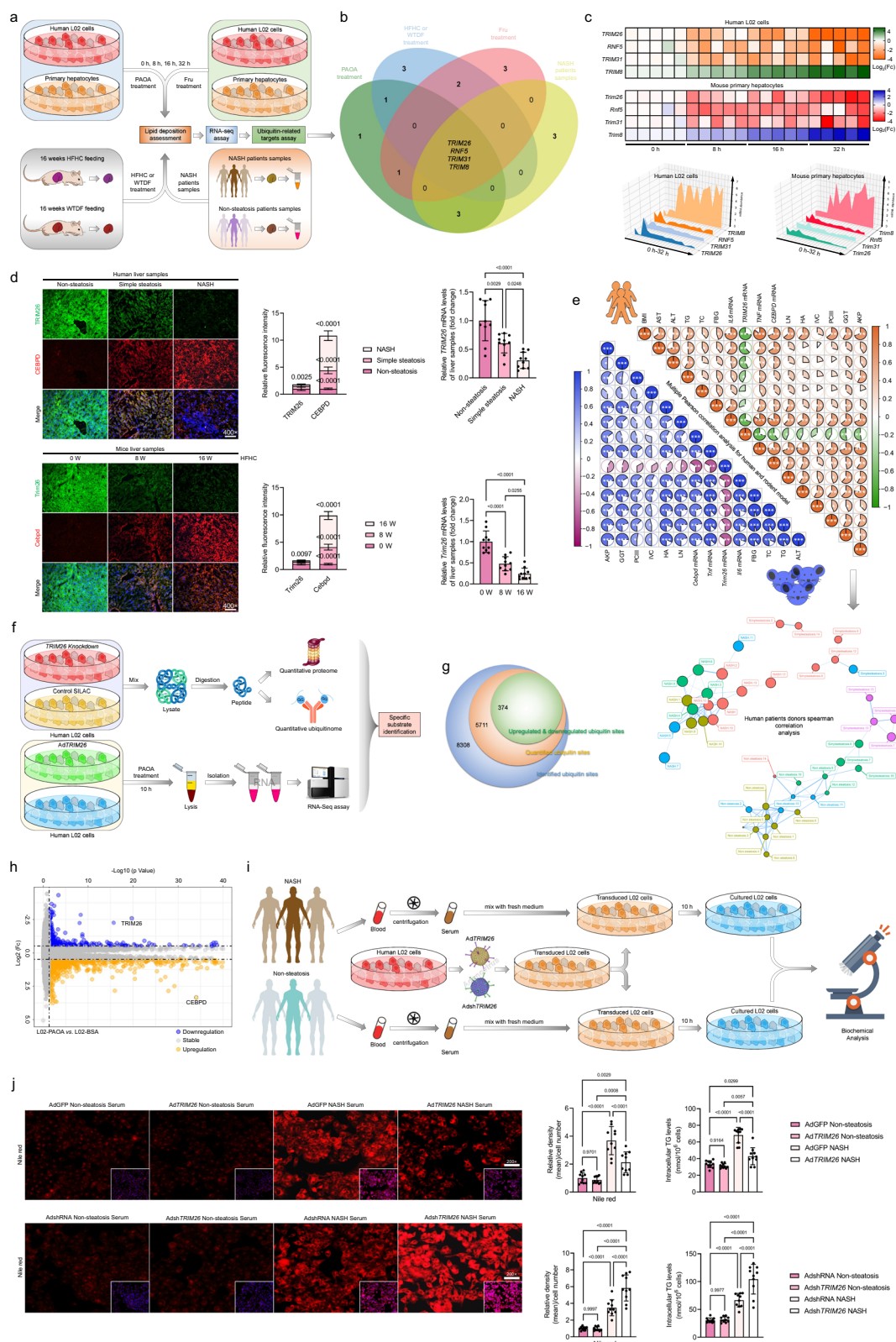

anticipated mutation in the binding location completely abolished the impact of HNF4A on the activation of the TRIM26 promoter. Additionally, our chromatin immunoprecipitation (ChIP) analysis provided further validation of the heightened enrichment levels of HNF4A at the specified loci within the TRIM26 promoter in HepG2 cells (Supplementary Fig. 1f). In both human liver samples and HFHC-treated mouse liver tissue, a significant positive connection between TRIM26 expression and HNF4A expression was consistently seen. This association was established using a two-tailed *t* test ($P < 0.001$) for human liver samples and a one-way ANOVA ($P < 0.05$) for the time course analysis of HFHC-treated mouse liver tissue (Supplementary Fig. 1g, h). In the present study, an additional multiple Pearson correlation analysis was conducted to examine the relationship between TRIM26 and NASH progression in both human and rodent models. The findings of

**Fig. 1 | TRIM26 expression is downregulated in fatty liver. a** Experimental design showing the protocol of identifying ubiquitin-related targets in response to a time-course of 0.5 mM palmitate+1.0 mM oleic acid (PAOA), 5 mM fructose (Fru) in indicated groups. **b** Venn diagram showing the Top 4 distinguishable expressed ubiquitin-related ligases candidates in intersection of indicated groups. **c** Heatmaps determining the 4 distinguishable expressed ubiquitin-related ligases indicators in human L02 cells and primary hepatocytes with PAOA treatment for 0–32 h. **d** Representative immunofluorescence images of Tripartite motif containing 26 (TRIM26) and CCAAT/enhancer binding protein delta (C/EBPD) co-expression in lives samples with fluorescence intensity evaluation (scale bars: magnification, ×400, $n = 10$ samples) and (high-fat+high-cholesterol) HFHC-fed mice, and corresponding liver *TRIM26* mRNA expression profiles ($n = 10$ samples per group) ($P < 0.0001$ by one-way ANOVA). **e** Multiple Pearson multiple correlation analysis for human subjects and rodent model (upper), and human patients donors network Spearman correlation analysis (lower) exhibiting the comprehensive correlation between *TRIM26* mRNA expression and indicated parameter indexes ($n = 49$ indices per parameter). **f** The flowchart showing the experimental procedure for the quantified ubiquitome, transcriptome and protein interaction assay of TRIM26. **g** Number of identified ubiquitin sites, upregulated & downregulated ubiquitin sites. **h** Volcano plot showing genes expression variation in L02 cells after PAOA treatment. Average protein expression ratio of 3 replicates (log 2 transformed) between PAOA- or BSA-induced L02 cells was plotted against *p*-value by Student's *t* test (−log 10 transformed). Cutoff of $P < 0.05$ and 1.2-fold change were marked by black dotted lines, respectively. The figure shows the total number of proteins identified as well as the number of up- and downregulated proteins. **i** A schematic diagram showing the human L02 cells were transfected with adenovirus (Ad*TRIM26*) or (Adsh*TRIM26*) for 24 h and then treated with the NASH serum- or Non-steatosis serum-mixed medium for 10 h. **j** Representative images and intracellular triglyceride (TG) analysis of the Nile red staining of L02 cells in indicated groups (scale bars: magnification, 200×; $n = 10$ images per group; $P < 0.05$ by one-way ANOVA). Data are expressed as mean ± SEM. The relevant experiments presented in this part were performed independently at least three times. $P < 0.05$ indicates statistical significance.

this analysis shed insight on the involvement of TRIM26 in NASH advancement and its association with other markers associated with NASH. Notably, consistent with other results, TRIM26 in human patients and a rodent model was negatively correlated with CEBPD, liver function indicators, inflammation, and profibrosis-related indicators. The lower graph of human patients' spearman correlation analysis was to determine the correlation between non-steatosis, simple steatosis, and NASH groups based on TRIM26 expression abundance. Indeed, hepatic TRIM26 expression was also tightly correlated with the severity of NAFLD/NASH in 3 donor groups (Fig. 1e). Having a tight correlation between TRIM26 and NASH pathogenesis, according to the approach of previous reports[16,17], we performed mass spectrometry-based quantitative proteomic analysis of stable isotope-labelled amino acids in cell culture (SILAC). Of note, we further confirmed that in response to a 10 h PAOA challenge (Fig. 1f–h), or time course of PAOA, TNF-α or fructose treatment (Supplementary Fig. 2a–c), TRIM26 was significantly downregulated in vitro assay and highlighted its potential substrate. To investigate in depth and identify the major catalytic substrate of TRIM26 in NASH pathology, an RNA-seq assay was next used to analyse gene expression changes in L02 cells with adenovirus-mediated *TRIM26* overexpression (Ad*TRIM26*) after a 10 h PAOA challenge (Fig. 2a). Indeed, transfected L02 cells with PAOA co-treatment exhibited that TRIM26 affects lipid metabolism and inflammatory responses. Accordingly, CEBPD as a major target of TRIM26 was also determined by volcano plot (Fig. 2b), PCA analysis (Fig. 2c), heatmap (Fig. 2d), KEGG pathway & wiki-pathway analysis (Fig. 2e), Sankey diagram enrichment (Fig. 2f) and GO biological process analysis (Fig. 2g). Also, the circle diagram analysed by GeneMANIA further confirmed the TRIM26-CEBPD protein interaction network in *Homo sapiens* and *Mus musculus* (Fig. 2h).

Considering the dramatic changes in TRIM26 expression in fatty liver, we next created a specific in vitro experimental model using Ad*TRIM26* or shRNA-induced *TRIM26* knockdown (Adsh*TRIM26*) constructs to assess the molecular function of TRIM26 in regulating critical markers of lipid accumulation (Fig. 1i). Transfected L02 cells or primary hepatocytes were treated for uninterrupted 10 h with serum isolated from NASH donors (NASH serum) or non-steatosis donors (Non-steatosis serum) ($P < 0.05$ by one-way ANOVA; Fig. 1j) or PAOA administration, respectively ($P < 0.05$ by one-way ANOVA for Supplementary Fig. 2D; $P < 0.001$ by 2 tailed *t* test for Supplementary Fig. 2e, f; $P < 0.05$ by one-way ANOVA for Supplementary Fig. 2g; $P < 0.001$ by 2 tailed *t* test for Supplementary Fig. 1h, i; Supplementary Fig. 2a–i). Unsurprisingly, the intracellular TG detection and oil red O staining indicated that NASH serum-treated L02 cells lipid deposition in the Ad*TRIM26* groups was markedly suppressed compared to those in the non-steatotic serum groups and was followed by downregulated concentrations of TG. On the contrary, inhibition of TRIM26 promoted lipid accumulation in transfected L02 cells. Similar results were further observed in the Adsh*Trim26*-mediated increase of lipid deposition in primary hepatocytes under NASH-like conditions ($P < 0.01$ by one-way ANOVA for Supplementary Fig. 3b; $P < 0.001$ by 2 tailed *t* test for Supplementary Fig. 3c, d; $P < 0.05$ by one-way ANOVA for Supplementary Fig. 3f; $P < 0.05$ by 2 tailed *t* test for the Supplementary Fig. 3g, h; Supplementary Fig. 3a–h). These results reveal that TRIM26 plays a protective role during NASH progression.

**Hepatocyte-specific *Trim26* ablation facilitates HFHC- or WTDF-triggered NASH pathological phenotypes**

To thoroughly investigate the functional role of Trim26 in the setting of NASH in vivo, we then established hepatocyte-specific *Trim26*-deficient mice (HKO) (Supplementary Fig. 4a–d), followed by a 16-week HFHC diet or WTDF diet, respectively. Unsurprisingly, the HKO mice displayed higher liver weights and LW/BW ratios than those in HFHC-treated Flox mice ($P < 0.0001$ by one-way ANOVA; Fig. 3a, b). Meanwhile, the HKO mice displayed higher fasting blood glucose, fasting insulin, and its corresponding HOMA-IR index ($P < 0.001$ by one-way ANOVA; Fig. 3c–e) than those of the corresponding control mice. Also, the HFHC-treated HKO mice further had more severe liver steatosis than those in HFHC-treated Flox mice, as indicated by hepatic TC, NEFA, and TG levels ($P < 0.0001$ by one-way ANOVA; Fig. 3f), pearson correlation analysis (Fig. 3g, h), H&E staining ($P < 0.05$ by 2 tailed *t* test; Fig. 3i), and oil red O staining ($P = 0.0003$ by 2 tailed *t* test; Fig. 3j). Consistently, hepatic inflammation was also significantly elevated in HKO mice compared to Flox mice, as indicated by positive CD11b- ($P < 0.0001$ by 2 tailed *t* test) and F4/80-associated inflammatory cell infiltration ($P = 0.0004$ by 2 tailed *t* test) in liver samples (Fig. 3j). Additionally, according to sirius red staining ($P = 0.0001$ by 2 tailed *t* test) and masson staining ($P < 0.0001$ by 2 tailed *t* test) analysis, hepatic collagen deposition was higher in liver tissue from HFHC-administrated HKO mice than in corresponding Flox mice (Fig. 3j). Also, the serum contents of pro-inflammatory cytokines and gene expression profiles exhibited alterations in inflammation ($P < 0.05$ by 2 tailed *t* test; Fig. 3k), fatty acid metabolism ($P < 0.05$ by 2 tailed *t* test; Fig. 3l), profibrotic genes ($P < 0.05$ by 2 tailed *t* test; Fig. 3m) and hepatic function indicators (e.g., ALT, AST, AKP and GGT) ($P < 0.05$ by 2 tailed *t* test; Fig. 3n) were markedly higher in the HFHC-induced HKO mice than those in the controls.

The aetiology of NASH is intricate and recognised as a multi-faceted condition, characterised by a diverse range of factors influenced by both external environmental elements and individual genetic predispositions. These aspects are challenging to adequately duplicate in animal models, further complicating the study of NASH pathogenesis[17]. Therefore, we further investigated Trim26 function on a WTDF diet supplemented with a 15% w/v fructose-drinking

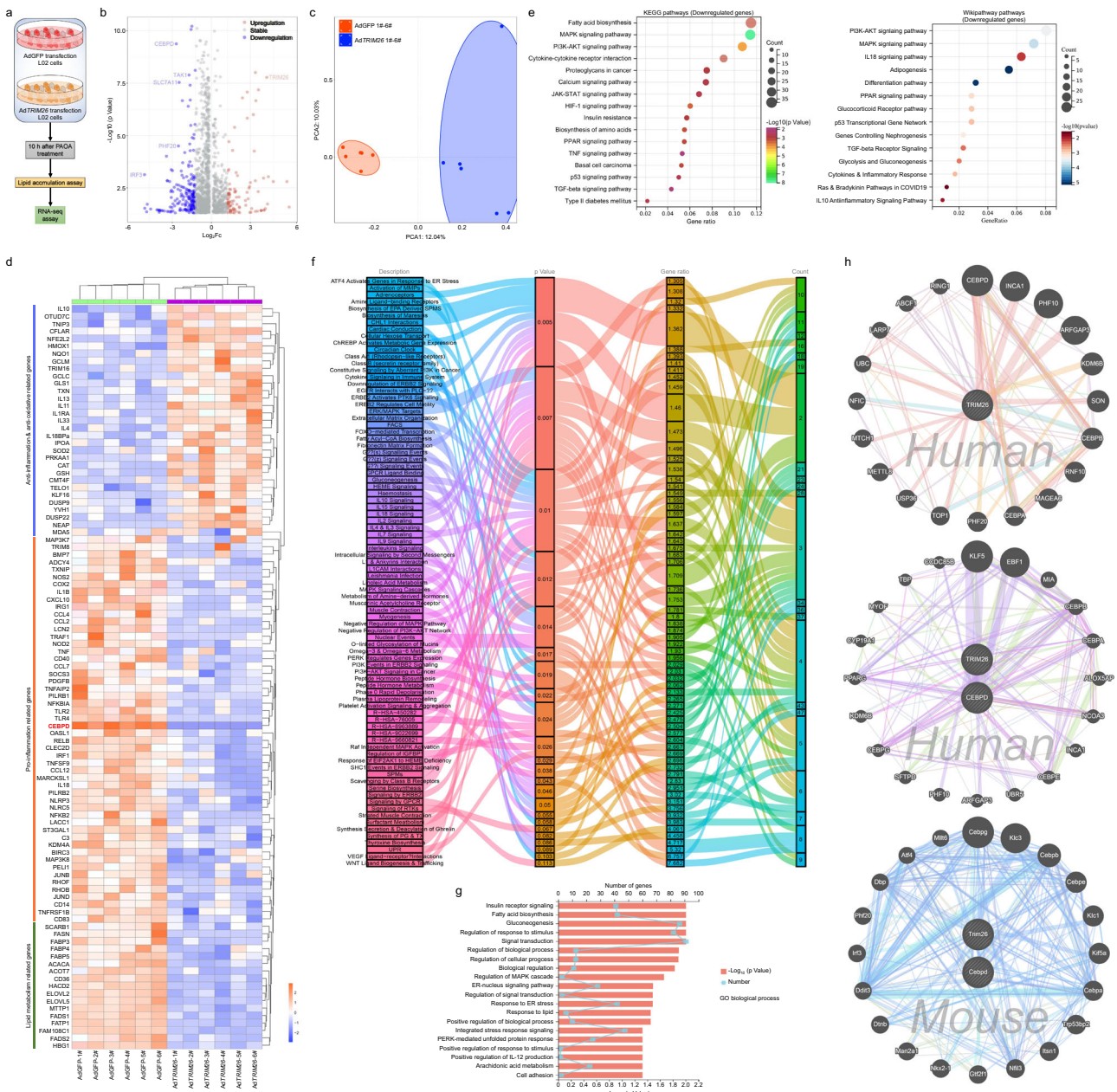

**Fig. 2 | Identification of CEBPD as a target of TRIM26. a** Schematic of the experimental procedure showing the genes expression in L02 cells after adenovirus-mediated *TRIM26* overactivation (Ad*TRIM26*). **b–g** Volcano plot; average protein expression ratio of 3 replicates (log 2 transformed) between 0.5 mM palmitate+1.0 mM oleic acid (PAOA)-induced AdGFP- or Ad*TRIM26*-transfected L02 cells was plotted against *p*-value by Student's *t* test (−log 10 transformed). Cutoff of *P* < 0.05 and 1.2-fold change were marked by black dotted lines, respectively. The figure shows the total number of proteins identified as well as the number of up-

and downregulated proteins (**b**), principal component analysis (PCA) analysis (**c**), heatmap (**d**), KEGG pathway & wiki-pathway analysis (**e**), Sankey diagram enrichment (**f**) and GO biological process analysis; data was plotted against *p*-value by Student's *t* test (−log 10 transformed) with cutoff of *P* < 0.05 and 1.2-fold change (**g**) in AdGFP-transfected or Ad*TRIM26*-transfected L02 cells with PAOA co-treatment showing that TRIM26 affects lipid metabolism and inflammatory responses. **h** Circle diagram analysed by GeneMANIA further showing TRIM26-CEBPD protein interaction network in *Homo sapiens* and *Mus musculus*.

water-induced NASH model that simulates pathogenesis. Consistent with the results obtained in HFHC-induced HKO mice, indeed, the HKO mice cannot render a significant alteration in body weight but exhibit higher liver weight (*P* = 0.5917 by 2 tailed *t* test) and LW/BW ratio (*P* = 0.0246 by 2 tailed *t* test) in response to 16-week WTDF administration (Supplementary Fig. 5a, b). In addition, the fasting blood glucose (*P* = 0.0248 by 2 tailed *t* test), fasting insulin contents (*P* < 0.0001 by 2 tailed *t* test), HOMA-IR index (*P* < 0.0001 by 2 tailed *t* test), hepatic lipid contents (i.e., TG, NEFA, and TC) (*P* < 0.001 by 2 tailed *t* test), as well as the NAS score (*P* = 0.0202 by 2 tailed *t* test), inflammation

(*P* < 0.01 by 2 tailed *t* test), collagen deposition (*P* < 0.001 by 2 tailed *t* test) and hepatic function indicators (i.e., ALT, AST, AKP, and GGT) (*P* < 0.01 by 2 tailed *t* test) (Supplementary Fig. 5c–j) were also markedly elevated in the HKO mice compared to Flox mice. Of note, to better study the protective function of Trim26 in the NASH in vivo model, the ex vivo gene therapy strategy of lentivirus-loaded full-length *Trim26* sequences (LV-*Trim26*) or shRNA targeting *Trim26* (LV-sh*Trim26*) was used to further explore the role of Trim26 in the alleviation of steatohepatitis (Supplementary Fig. 6a–c). As expected, after ex vivo transplantation, the HFHC-fed mice with hepatocyte-specific

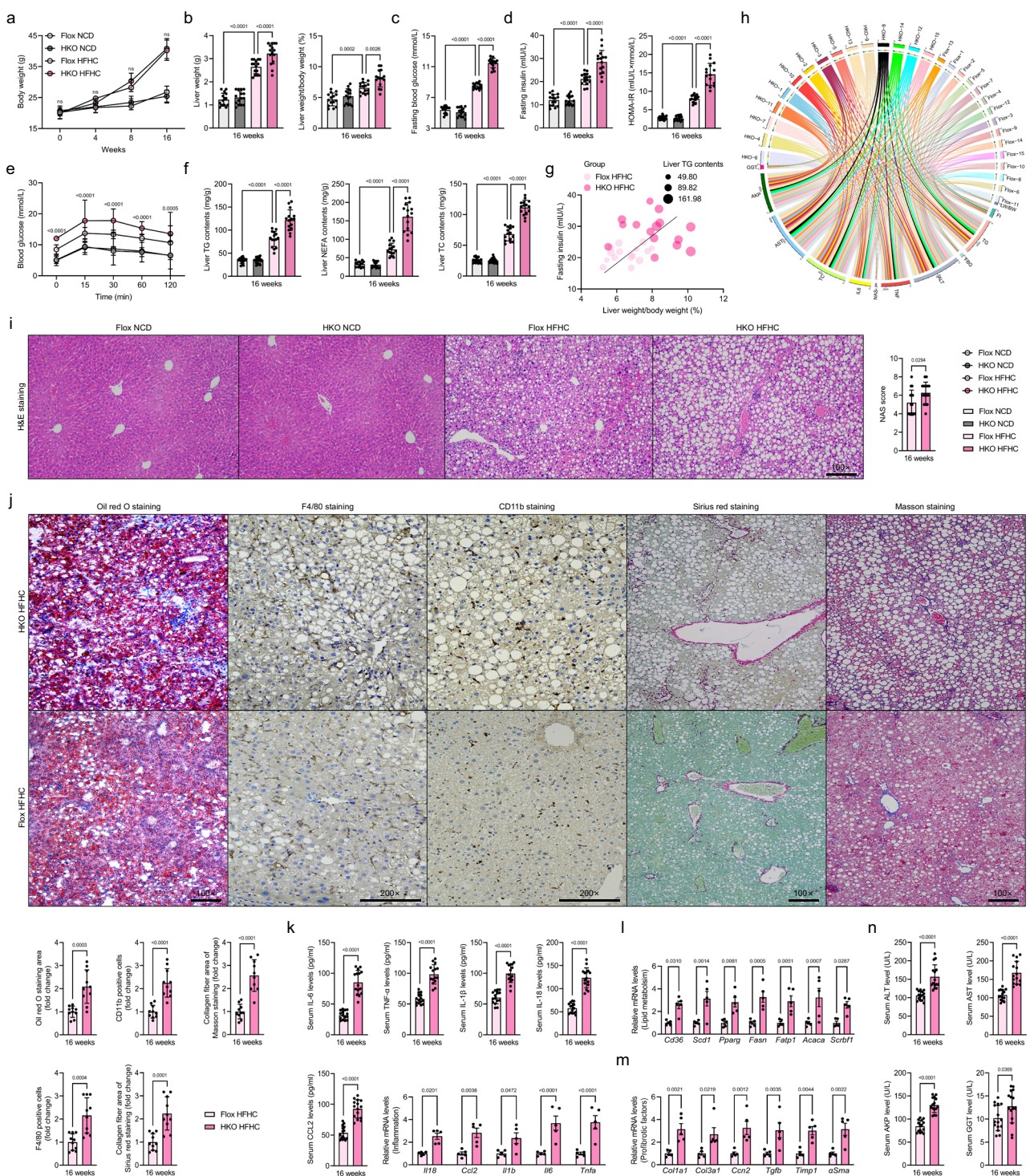

*Trim26* knockdown (HFHC LV-sh*Trim26*) lead to a remarkable increase in liver weight ($P = 0.0005$ by 2 tailed *t* test), LW/BW ratio ($P = 0.0005$ by 2 tailed *t* test), fasting insulin ($P < 0.0001$ by 2 tailed *t* test), fasting blood glucose ($P = 0.0153$ by 2 tailed *t* test), corresponding HOMA-IR index ($P < 0.0001$ by 2 tailed *t* test), liver TG, TC, and NEFA contents ($P < 0.05$ by 2 tailed *t* test), as compared to those of controls (HFHC LV-shRNA) (Supplementary Fig. 7a–c). Moreover, HFHC LV-sh*Trim26* mice had more increased hepatosteatosis and inflammation in their livers than HFHC LV-shRNA mice, as indicated by the H&E staining, NAS score ($P = 0.0121$ by 2 tailed *t* test), infiltration of F4/80 positive inflammatory cells ($P < 0.0001$ by 2 tailed *t* test), aberrant gene expression involving lipid uptake, synthesis, and collagen deposition,

upregulated liver function indicators, increased pro-inflammatory cytokines (e.g., IL-6, TNF-α, IL-1β, IL-18, and CCL2), and correspondingly reduced anti-inflammatory factor (i.e., IL-10) ($P < 0.05$ by 2 tailed *t* test) (Supplementary Fig. 7d–h). Additionally, functional loss of hepatocyte *Trim26* accelerates HFHC/WTDF-triggered NASH-associated fibrosis pathologies, as evidenced by increased liver α-Sma contents, serum TGF-β, hepatic hydroxyproline levels, corresponding pro-fibrotic factors, e.g., *α-Sma*, *Tgfb*, *Ctgf*, *Col1a1*, *Col3a1* mRNA abundance, and pro-fibrotic factors levels in the HKO HFHC/WTDF-Serum-treated primary HSCs assay ($P < 0.05$ by 2 tailed *t* test; Supplementary Fig. 8a–h). In light of the tight relation between fat accumulation and the development of metabolic disorders, there is

**Fig. 3 | Hepatocyte-specific *Trim26* deficiency exacerbates HFHC-induced NASH pathologies. a–f** Records for the body weight (*n* = 15 per group; *ns.*, no significance) (**a**), liver weight and the ratio of liver weight/body weight (%) (LW/BW) (*n* = 15 per group; *P* < 0.001 by one-way ANOVA) (**b**), fasting blood glucose levels (*n* = 15 per group; *P* < 0.0001 by one-way ANOVA) (**c**), fasting insulin levels, homoeostasis model assessment of insulin resistance (HOMA-IR) index (*n* = 15 per group; *P* < 0.0001 by one-way ANOVA) (**d**), glucose tolerance test (GTT) (*n* = 15 per group; *P* < 0.001 by one-way ANOVA) (**e**) and liver triglyceride (TG), non-esterified fatty acid (NEFA) and total cholesterol (TC) contents (**f**) of the indicated groups (*n* = 15 per group; *P* < 0.001 by one-way ANOVA). **g, h** Pearson analysis and circle correlation indicating the correlations between fasting insulin and ratio of LW/BW (**g**), and different indicators correlation (**h**) in HFHC-fed HKO or Flox mice. Adjustments to data were made for multiple comparisons. *P* < 0.001 for all of these correlations (*n* = 15 per parameter). Correlations were performed using Pearson rank correlation coefficient analysis. **i, j** Representative pictures of H&E staining, histological NAS score (**i**) changes and oil red O staining (*P* = 0.0003 by 2 *t* test), F4/ 80 (*P* = 0.0004 by 2 *t* test) and CD11b (*P* < 0.0001 by 2 *t* test) expression, masson staining (*P* < 0.0001 by 2 *t* test) and sirius red staining (*P* = 0.0001 by 2 *t* test) (**j**) in HFHC-fed HKO or Flox mice (scale bars: magnification, ×100 for H&E staining, oil red O staining, masson staining and sirius red staining; scale bars: magnification, ×200 for F4/80 and CD11b staining; *n* = 10 images per group). **k** Representative inflammation-related cytokines and genes expression profiles (*P* < 0.0001 by 2 tailed *t* test) in serum (*n* = 15 per group) or livers from HFHC-fed HKO or Flox mice (*n* = 5 per group). (**l, m**) Representative mRNA levels of fatty acid metabolism (**l**)- and profibrotic factors (**m**)-associated genes expression in indicated liver tissue (*n* = 5 per group; *P* < 0.05 by 2 *t* test). **n** Representative liver function-related indicators including ALT (*P* < 0.0001 by 2 tailed *t* test), AST (*P* < 0.0001 by 2 tailed *t* test), AKP (*P* < 0.0001 by 2 tailed *t* test) and GGT (*P* = 0.0369 by 2 tailed *t* test) in serum from HFHC-fed HKO or Flox mice (*n* = 15 mice per group). Data are expressed as mean ± SEM. The relevant experiments presented in this part were performed independently at least three times. *P* < 0.05 indicates statistical significance.

indeed, no significant difference in food intake between NCD- and HFHC/WTDF-induced HKO mice (*P* < 0.05 by 2 tailed *t* test; Supplementary Fig. 9a, b). However, the raising of visceral fat weight was facilitated in the HFHC/WTDF-induced HKO mice, accompanied by a significant upregulation in the ratio of visceral fat weight to body weight (*P* < 0.05 by 2 tailed *t* test; Supplementary Fig. 9c, d). Furthermore, the adipose cell size was markedly enlarged between NCD- and HFHC/WTDF-induced HKO mice, but there was no statistical difference between HFHC/WTDF-induced HKO mice and corresponding Flox mice (*P* < 0.05 by 2 tailed *t* test; Supplementary Fig. 9e, f). Collectively, these findings indicated that liver functional loss of *Trim26* worsens hepatic lipid dysregulation, inflammation, and collagen deposition, resulting in steatohepatitis and early hepatofibrosis in mice.

## Hepatocyte-specific *Trim26* overexpression retards HFHC- or WTDF-triggered NASH pathological phenotypes

Given the inverse relevance of Trim26 expression with severity of NASH in mice models with *Trim26* loss-of-function experiments based on Rosa26 conditional and/or inducible transgenesis (hereunder named as Rosa26$^{Trim26}$) (*P* < 0.0001 by 2 tailed *t* test; Supplementary Fig. 10a, b). The Rosa26$^{Trim26}$ mice injected with AAV8-TBG-Cre were used to specifically overexpress Trim26 in hepatocytes, followed by a 16-week HFHC or WTDF diet challenge (Fig. 4a and Supplementary Fig. 11a) (HTG HFHC or HTG WTDF). As expected, in contrast to the HKO mice, compared with HFHC-treated NTG mice, the liver weight and LW/BW of the HFHC-treated HTG mice were significantly reduced; however, there was no statistically significant difference in body weight between the two groups (*P* < 0.0001 by 2 tailed *t* test; Fig. 4b, c). Moreover, increased insulin levels, fasting blood glucose levels, the HOMA-IR index, hepatic TG, NEFA, and TC contents, hepatosteatosis, inflammation, and fibrosis-related indicators were downregulated by hepatic Trim26 overexpression (*P* < 0.01 by 2 tailed *t* test; Fig. 4d–o). Coincidentally, decreased dysregulated liver metabolism, lipid deposition, hepatosteatosis, and profibrosis, followed by reduced pro-inflammatory cytokines (e.g., IL-6, TNF-α, IL-1β, IL-18, and CCL2), and corresponding elevated anti-inflammatory factors (i.e., IL-10) were further observed by Trim26 overexpression in the WTDF-triggered NASH phenotype (*P* < 0.05 by 2 tailed *t* test; Supplementary Fig. S11b–k). Also, after ex vivo-mediated allogeneic hepatocyte transplantation, the HFHC-fed mice with LV-*Trim26* gene therapy exhibited a marked decrease in liver weight, LW/BW ratio, fasting blood glucose, fasting insulin, HOMA-IR index, and hepatic lipid contents (*P* < 0.05 by 2 tailed *t* test; Supplementary Fig. S12a–d), as compared to those of controls (HFHC LV-Control). Also, HFHC-fed LV-*Trim26* mice presented less liver steatosis than controls, as confirmed by the H&E staining, F4/80 staining, reduced NAS score, downregulated pro-inflammatory cytokines (e.g., IL-6, TNF-α, IL-1β, IL-18 and CCL2), and upregulated anti-inflammatory

factor (i.e., IL-10) (*P* < 0.05 by 2 tailed *t* test; Supplementary Fig. S12e, f). In addition, HFHC LV-*Trim26* mice showed a significant reduction in fatty acid uptake- and synthesis-, profibrosis- and proinflammation-associated gene expression, and abnormal liver function indicators (*P* < 0.01 by 2 tailed *t* test; Supplementary Fig. S12g–i) compared to the HFHC LV-Control mice. These results were also supported by decreased hepatic α-Sma levels, serum TGFβ, serum hydroxyproline, and another in vitro assay (*P* < 0.05 by 2 tailed *t* test; Supplementary Fig. 13a–h). Considering the close correlation between fat deposition and the development of metabolic syndrome, there is no marked statistical difference in food intake between NTG- and HFHC/WTDF-induced HTG mice. Nevertheless, the increase in visceral fat weight was restrained in the HFHC/WTDF-induced HTG mice, followed by a significant downregulation in the ratio of visceral fat weight to body weight (*P* < 0.05 by 2 tailed *t* test; Supplementary Fig. 14a–f). In short, these results suggest that overexpression of Trim26 retards liver lipid dysregulation, inflammation, and collagen accumulation, helping to mitigate steatohepatitis and early hepatofibrosis in mice.

## TRIM26 retards NASH progression by repressing CEBPD-HIF1A signalling

Given the significant inhibitory effects of Trim26 on the advancement of NASH and its related pathological processes, the aforementioned findings have compelled us to investigate the molecular mechanisms underlying Trim26 and its intrinsic functionality. In light of the determination of Cebpd as a potential target and substrate of Trim26, we next generated HKO (*Cebpd*) and hepatocyte-specific *Trim26* and *Cebpd* double deletion mice (DHKO (*Trim26-Cebpd*)) (Supplementary Fig. 15a). The conjoint analysis of mass spectrometry suggested that CEBPD is a partner protein and a direct downstream target of TRIM26 (*P* < 0.0001 by one-way ANOVA; Supplementary Fig. 15b, c). Furthermore, as expected, *Cebpd* deletion restrained the *Trim26* knockout-triggered upregulation of Cebpd signalling and pathogenetic progression in the context of the HFHC-induced NASH phenotype. It is worth mentioning that the various phenotypes associated with NASH that were observed in the context of *Trim26* deficiency, such as increased liver weight, abnormal glucose metabolism, liver lipid accumulation, elevated levels of proinflammatory cytokines in the liver, altered expression profiles of genes related to inflammation, deposition of collagen fibres in the liver, and hepatocellular injury, were significantly mitigated when *Cebpd* expression was deficient (*P* < 0.0001 by one-way ANOVA; Fig. 5a–k and Supplementary Fig. 15d–f). Besides, *Trim26* ablation did slightly elevate Cebpd-Hif1a signalling and its corresponding downstream pathways, including Cebpd, Hif1a, Nos2, p-p38, and p-p65 protein, in HFHC- and WTDF-induced fatty liver and NASH serum-treated Ad*TRIM26*- or Adsh*TRIM26*-transfected L02 cells (Fig. 6a–d). Moreover, the primary hepatocytes and L02 cells transfected or co-transfected with or

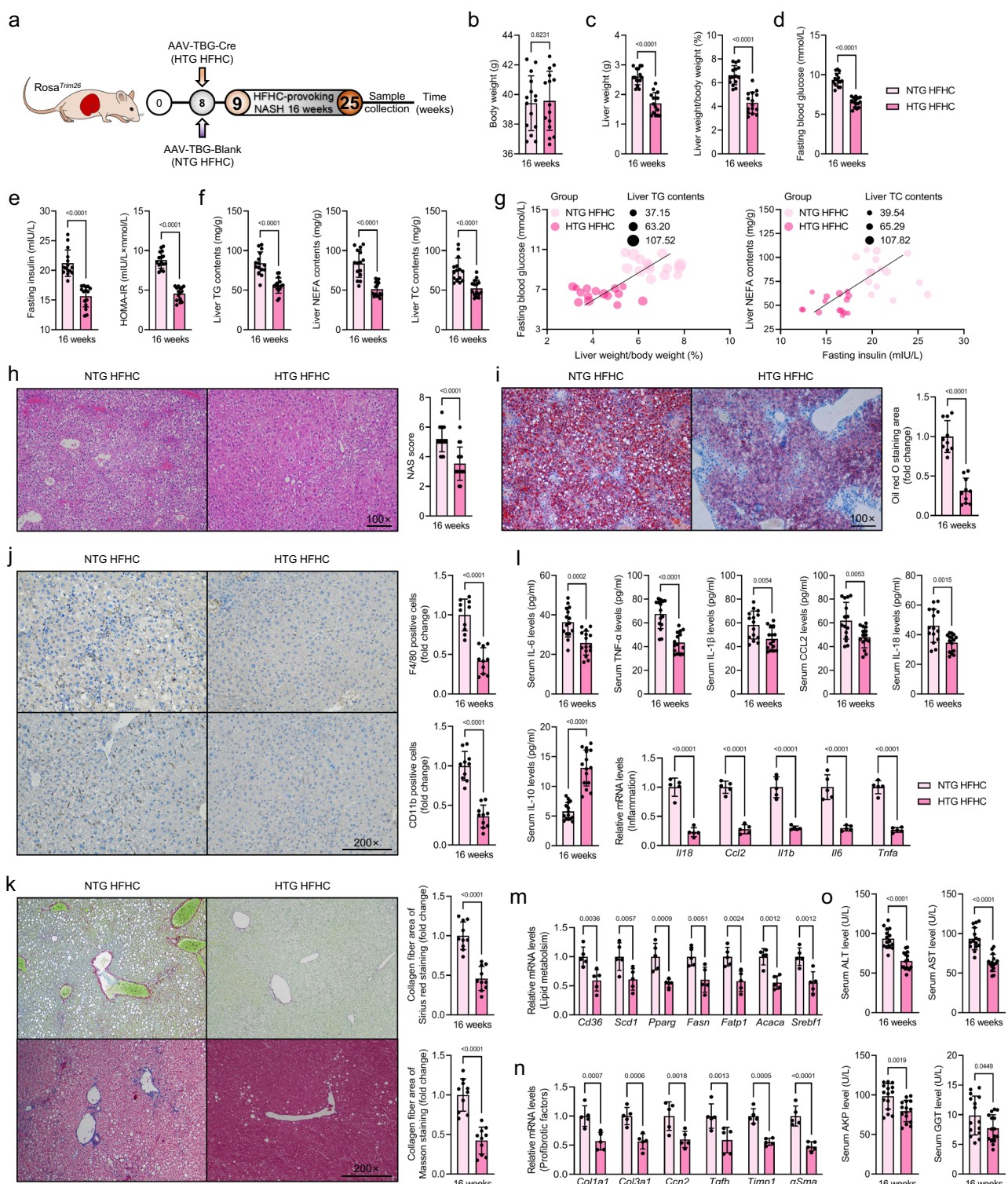

without Adsh*TRIM26*, Adsh*CEBPD*, or Adsh*TRIM26*/Adsh*CEBPD* also exhibited higher lipid deposition during PAOA treatment in vitro assay (*P* < 0.0001 by one-way ANOVA; Fig. 6e). Given the inhibitory role of TRIM26 in regulating CEBPD-associated signalling, the in vitro and in vivo results compelled us to detect another important concern: whether and how TRIM26 binds to CEBPD in the context of NASH progression. Inevitably, the results of the co-immunoprecipitation assay and GST pull-down experiments demonstrated that the exogenous expression of TRIM26 in transfected L02 cells led to an interaction with CEBPD, and vice versa (Fig. 6f). It is worth mentioning that the findings of the study revealed that the CC-PRY/SPRY (CPS) domain

of TRIM26 played a crucial role in its interaction with CEBPD. The study of Co-IP, in conjunction with the use of CEBPD mutants, provided additional evidence that the interaction between CEBPD and TRIM26 is dependent on the presence of the BR domain in CEBPD. Therefore, the BR domain of CEBPD had a role in the protein's interaction with TRIM26 (Fig. 6g). In addition, we have identified the amino acid sequences of the RING domain of TRIM26 in multiple species and have highlighted two highly conserved motifs, TRIM26 (CXXC) and TRIM26 (CXXXXC), within the RING domain (Fig. 6h). This finding forced us to study the role of the two motifs in regulating PAOA-mediated disturbances of lipid metabolism and inflammation in vitro. Sure enough,

**Fig. 4 | Hepatocyte-specific *Trim26* overexpression mitigates HFHC-induced NASH pathologies. a** Schematic diagram of construction of HTG HFHC mice model. **b**–**f** Records for the body weight ($P < 0.0001$ by 2 tailed *t* test) (**b**), liver weight and the ratio of liver weight/body weight (%) (LW/BW) ($P < 0.0001$ by 2 tailed *t* test) (**c**), fasting blood glucose levels ($P < 0.0001$ by 2 tailed *t* test) (**d**), fasting insulin levels and homoeostasis model assessment of insulin resistance (HOMA-IR) index ($P < 0.0001$ by 2 tailed *t* test) (**e**) and liver triglyceride (TG), non-esterified fatty acid (NEFA) and total cholesterol (TC) ($P < 0.0001$ by 2 tailed *t* test) (**f**) in indicated groups ($n = 15$ per group). **g** Pearson correlation analyses indicating the correlations between fasting blood glucose and ratio of LW/BW, and liver NEFA contents and fasting insulin levels in indicated groups ($n = 15$ per parameter; $n = 45$ in total). **h, i** Representative pictures of H&E staining, NAS score ($P < 0.0001$ by 2 tailed *t* test) (**h**) and oil red O staining ($P < 0.0001$ by 2 tailed *t* test) (**i**) in indicated groups (scale bars: magnification, ×100; $n = 10$ per group). (**j, k**) Immunohistochemical assay showing the F4/80 or CD11b expression (scale bars: magnification, ×200) ($P < 0.0001$ by 2 tailed *t* test) (**j**) and masson staining and sirius red staining (scale bars: magnification, 100×) ($P < 0.0001$ by 2 tailed *t* test) (**k**) -indicated liver histopathologic changes in liver of indicated groups ($n = 10$ per group). **l** Representative inflammation-related cytokines and genes expression profiles including interleukin-6 (IL-6) ($P = 0.0002$ by 2 tailed *t* test), tumour necrosis factor-α (TNF-α) ($P < 0.0001$ by 2 tailed *t* test), interleukin-1β (IL-1β) ($P = 0.0054$ by 2 tailed *t* test), interleukin-18 (IL-18) ($P = 0.0015$ by 2 tailed *t* test), chemokine CCL2 ($P = 0.0053$ by 2 tailed *t* test) and interleukin-10 (IL-10) ($P < 0.0001$ by 2 tailed *t* test) in serum ($n = 15$ mice per group) or livers ($n = 5$ per group; $P < 0.0001$ by 2 tailed *t* test) from indicated groups. **m, n** Representative mRNA levels of fatty acid metabolism ($P < 0.01$ by 2 tailed *t* test) (**m**)- and profibrotic factors ($P < 0.01$ by 2 tailed *t* test) (**n**)-associated genes expression in livers from indicated groups ($n = 5$ per group). **o** Representative liver function-related indicators including alanine transaminase (ALT) ($P < 0.0001$ by 2 tailed *t* test), aspartate aminotransferase (AST) ($P < 0.0001$ by 2 tailed *t* test), alkline phosphatase (AKP) ($P = 0.0019$ by 2 tailed *t* test) and glutamyl transpeptidase (GGT) ($P = 0.0449$ by 2 tailed *t* test) in serum from indicated groups ($n = 15$ per group). Data are expressed as mean ± SEM. The relevant experiments presented in this part were performed independently at least three times. $P < 0.05$ indicates statistical significance.

elevated lipid deposition and activated CEBPD-HIF1A signalling were not mitigated in Ad*TRIM26* (AXXA)- and Ad*TRIM26* (AXXXXA)-transfected L02 cells compared to corresponding AdGFP controls, but were inhibited in Ad*TRIM26*-transfected L02 cells during PAOA challenge ($P < 0.05$ by one-way ANOVA; Fig. 6i–k).

Considering the close relationship between CEBPD deactivation and catalysis function-associated RING domain activity of TRIM26 (including 2 conserved motifs), to further confirm whether TRIM26 with RING domain mutants are required for the inhibitory role of TRIM26 against NASH progression, we then subjected AAV-TBG-*Trim26*, AAV-TBG-*Trim26* (△RING), AAV-TBG-*Trim26* (AXXA) or AAV-TBG-*Trim26* (AXXXXA) to 16-weeks WTDF-triggered NASH model in vivo (Supplementary Fig. 16a and Supplementary Fig. 17a). Unsurprisingly, except for the mice injected with AAV-TBG-*Trim26*, groups injected with AAV-TBG-*Trim26* (△RING), E3 defective mutants, were not capable of assuaging liver lipid deposition, inflammation response, early fibrosis development, or hepatocellular injury after prolonged WTDF challenge compared to those of corresponding controls ($P < 0.05$ by one-way ANOVA; Supplementary Fig. 16b-k). Consistent with findings from AAV-TBG-*Trim26* (△RING) in vivo and in vitro, mice groups injected with AAV-TBG-*Trim26* (AXXA) or AAV-TBG-*Trim26* (AXXXXA) also failed to mitigate NASH progression ($P < 0.05$ by one-way ANOVA; Supplementary Fig. 17b–j). These additional results further revealed that two conserved motifs (Trim26 (AXXA) and Trim26 (AXXXXA)) of the RING domain in Trim26 are required for their inhibitory effects on the regulation of NASH pathogenesis. These results indicated that the RING domain of TRIM26 is required for its catalytic function.

## TRIM26 activity leads to proteasome degradation of CEBPD

Previous studies have indicated that CEBPD is ubiquitinated[18], and upregulation of CEBPD induced by a variety of stimuli significantly promoted inflammation and hepatic lipogenesis[19–21]. Indeed, a markedly increased abundance of CEBPD was observed in the livers of NASH patients and HFHC/WTDF diet-treated rodent models as compared to those of non-steatosis donors and NCD diet-fed control mice (Fig. 1d). In our current work, TRIM26, an important member of the E3 ubiquitin ligases, has been identified as a CEBPD-associated protein and a potential suppressor of CEBPD. In light of these findings, we speculated whether TRIM26 decreased CEBPD abundance by facilitating proteasome degradation of CEBPD. Accordingly, we detected the effect of activated TRIM26 on CEBPD protein stability. Suppression of protein synthesis by cycloheximide (CHX) in transfected L02 cells confirmed that CEBPD was less stable when expressed in the presence of flag-tagged full-length TRIM26 (TRIM26-FL-Flag) (Fig. 7a, b). Increased CEBPD expression was subjected to this in vitro assay to have detectable protein at the start of the time frame with TRIM26. Of

note, many protein factors are turned over through proteasomal-dependent degradation. When cultured L02 cells or primary hepatocytes were incubated with the proteasome suppressor MG132, endogenous CEBPD expression abundance could be restrained in the presence of TRIM26 (Fig. 7c). Also, in the absence of TRIM26, the proteasome suppressor MG132 had no marked effect on CEBPD expression in these transfected cells. Since nearly all proteasome substrates are targeted for polyubiquitination-mediated degradation, we then investigated whether CEBPD was polyubiquitinated. As expected in Fig. 7d, the ubiquitin assay of CEBPD from transfected L02 cells expressing HA-tagged ubiquitin indicated that TRIM26 resulted in high-molecular-weight, polyubiquitinated CEBPD. In addition, given the unavailable catalytic function of TRIM26 (AXXA) and TRIM26 (AXXXXA) in regulating CEBPD-related NASH progression, we next assessed whether the two mutant motifs of the RING domain in TRIM26 directly affected the ubiquitination of CEBPD. Consistent with in vivo experiments, in *TRIM26*-deficient L02 cells or primary hepatocytes, only cultured cells with full-length TRIM26 restoration were capable of promoting the ubiquitination process of CEBPD, while cells transfected with TRIM26 (AXXA) or TRIM26 (AXXXXA) mutants failed to perform their catalytic function (Fig. 7e). The similar results were further supported by lipid deposition and TG contents assays in Ad*TRIM26*-, Ad*TRIM26* (AXXA)-, or Ad*TRIM26* (AXXXXA)-transfected primary hepatocytes after 10 h of PAOA challenge ($P < 0.0001$ by one-way ANOVA; Fig. 7f). Moreover, previous reports have demonstrated that polyubiquitination happens commonly on lysine residues[18]. For the CEBPD protein, the K120 residue has been confirmed as a major site of sumoylation and is thereby a candidate site for ubiquitination. Unsurprisingly, CEBPD with the K120A mutant blocked TRIM26-triggered polyubiquitination degradation of CEBPD (Fig. 7g) in transfected L02 cells, accompanied by failure of TRIM26-mediated decrease of lipid accumulation and TG contents in Ad*CEBPD*, Ad*TRIM26* + Ad*CEBPD*, or Ad*TRIM26* + Ad*CEBPD* (K120A) co-transfected L02 cells, respectively ($P < 0.01$ by one-way ANOVA; Fig. 7h). These above findings indicate that TRIM26-regulated CEBPD degradation via polyubiquitination relies on its ubiquitin ligase activity and is TRIM26-dependent.

## Forced hepatocyte Trim26 activation alleviates the HFMCD-induced rodent NASH phenotype and the HFHC-induced rabbit NASH phenotype

Given the distinct mechanisms underlying the progression of NASH caused by the HFMCD diet compared to other high-energy diets such as HFHC or WTDF, it became necessary for us to investigate whether Trim26 could also provide protection against the development of NASH in rodent models induced by the HFMCD diet, as well as in rabbit models induced by the HFHC diet. As a result, the Rosa$^{Trim26}$ mice were

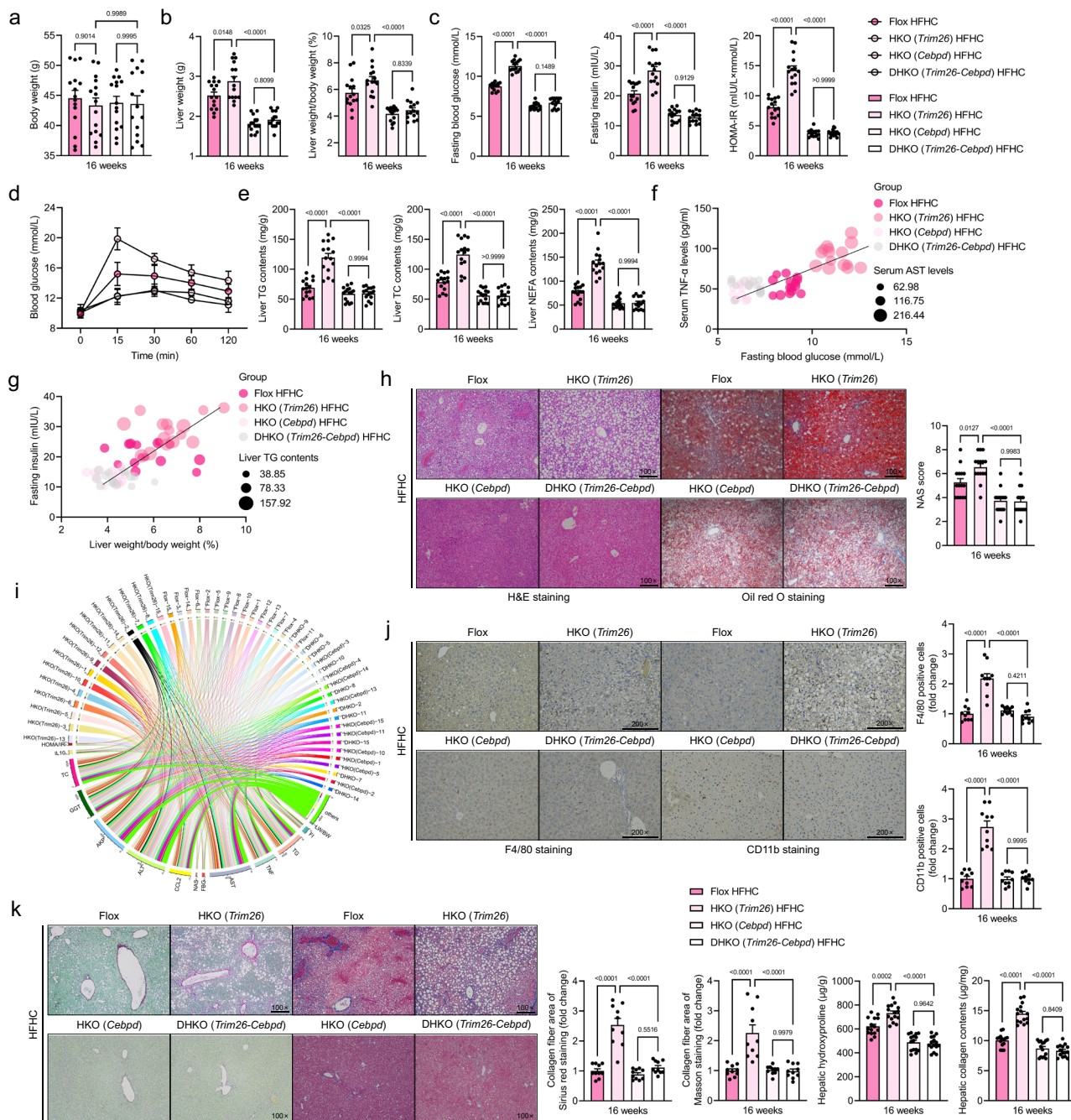

**Fig. 5 | Cebpd signalling is required for the protective function of Trim26 against NASH diet-induced steatohepatitis. a–d** Records for the body weight (**a**), liver weight and the ratio of liver weight/body weight (%) (LW/BW) (*P* < 0.0001 by one-way ANOVA) (**b**), fasting blood glucose levels, fasting insulin levels, fasting insulin levels and homoeostasis model assessment of insulin resistance (HOMA-IR) index (*P* < 0.0001 by one-way ANOVA) (**c**) and glucose tolerance test (GTT) analysis (*P* < 0.0001 by one-way ANOVA) (**d**) of the HFHC-fed HKO (*Trim26*) mice, hepatocyte-specific *Cebpd* deletion-HKO (*Cebpd*) mice, hepatocyte-specific *Trim26* and *Cebpd* double knockout-DHKO (*Trim26-Cebpd*) mice and Flox mice (*n* = 15 mice per group). **e** Liver lipid contents including triglyceride (TG) (*P* < 0.0001 by one-way ANOVA), total cholesterol (TC) (*P* < 0.0001 by 2 tailed *t* test) and non-esterified fatty acid (NEFA) (*P* < 0.0001 by one-way ANOVA) in the indicated group (*n* = 15 mice per group). **f**, **g** Pearson correlation analyses showing the correlations between serum tumour necrosis factor (TNF-α) levels and fasting blood glucose (**f**), and fasting insulin and ratio of liver weight/body weight (%) (**g**) in HKO (*Trim26*) HFHC, HKO (*Cebpd*) HFHC, DHKO (*Trim26-Cebpd*) HFHC and Flox HFHC group. *P* < 0.01 for all

of these correlations (*n* = 15 per parameter; *n* = 45 in total; *P* < 0.0001 by one-way ANOVA). **h** Representative pictures of H&E staining, oil red O staining and NAS score (*P* < 0.0001 by one-way ANOVA) in indicated group (scale bars: magnification, 100×; *n* = 15 images per group for each staining). **i** Circle correlation analysis showing the correlation between a series of indicators in HKO (*Trim26*) HFHC, HKO (*Cebpd*) HFHC, DHKO (*Trim26-Cebpd*) HFHC and Flox HFHC group. **j**, **k** Representative pictures of immunohistochemical analysis of F4/80 (*P* < 0.0001 by one-way ANOVA) and CD11b (*P* < 0.0001 by one-way ANOVA) (scale bars: magnification, ×200; *n* = 15 images per group for each staining) (**j**) and sirius red (*P* < 0.0001 by one-way ANOVA) & masson staining (*P* < 0.0001 by one-way ANOVA) (scale bars: magnification, ×100; *n* = 15 images per group for each staining) (**k**)-indicated histological changes in liver, and corresponding hepatic hydroxyproline & collagen contents in HKO (*Trim26*) HFHC, HKO (*Cebpd*) HFHC, DHKO (*Trim26-Cebpd*) HFHC and Flox HFHC group. Data are expressed as mean ± SEM. The relevant experiments presented in this part were performed independently at least three times. *P* < 0.05 indicates statistical significance.

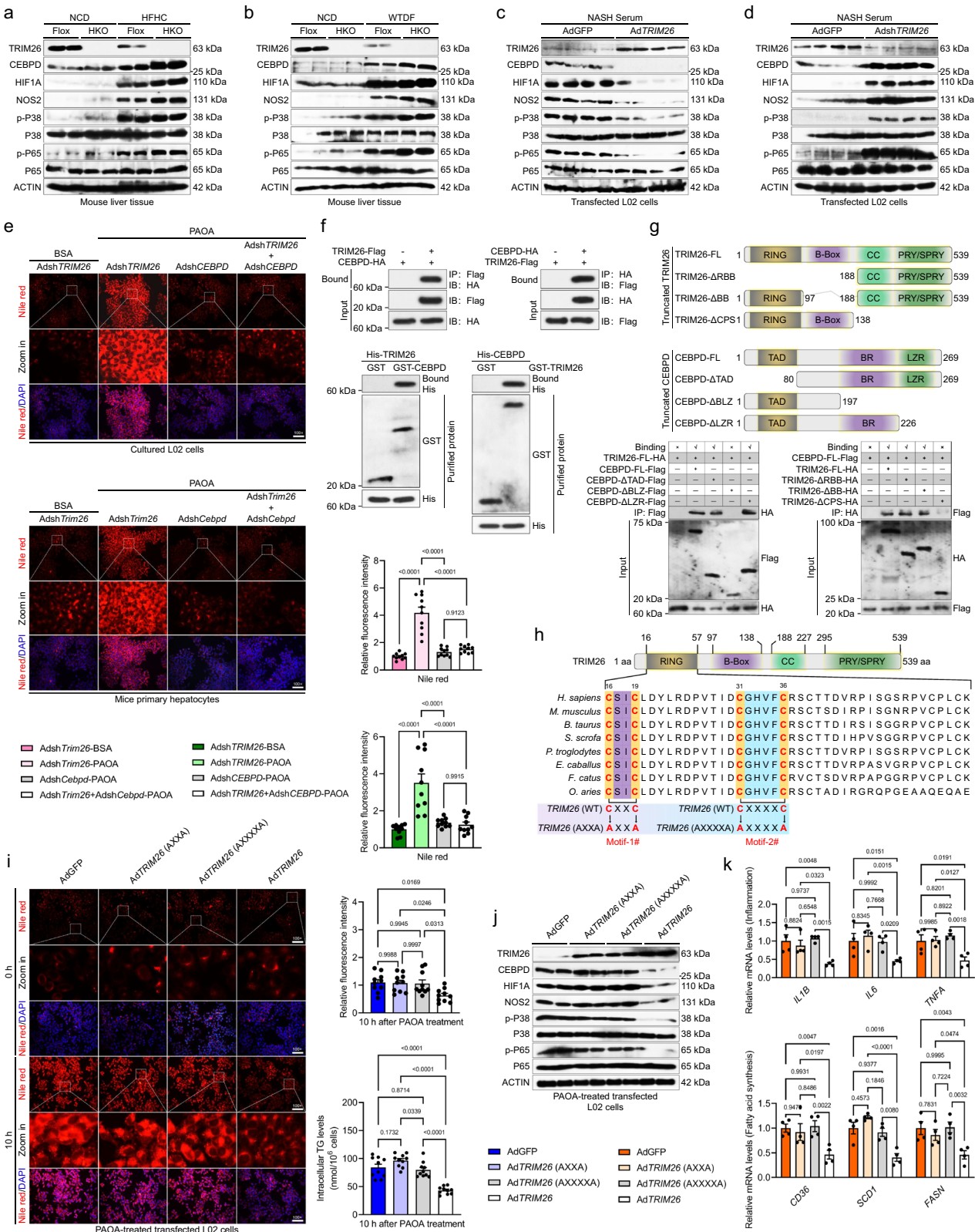

administered AAV-TBG-Cre vectors to induce an overexpression of Trim26 specifically in hepatocytes. Subsequently, these mice were subjected to an 8-week diet consisting of a HFMCD regimen (Fig. 8a), referred to as HTG HFMCD in subsequent discussions. The control group in this study consisted of Rosa^Trim26 mice that received an injection of AAV-TBG-Blank vector, and they will be referred to as NTG HFMCD mice for the remainder of this report. Indeed, the feeding of

HFMCD diet not only immediately initiates the pathogenesis of NASH through lipid toxicity, but also induces significant weight loss in mice. However, weight loss in HTG mice upon HFMCD administration was significantly rescued and later improved by Trim26 overexpression ($P < 0.05$ by one-way ANOVA; Fig. 8b). Meanwhile, HTG mice also exhibited decreased hepatosteatosis than they did in NTG mice after an 8-week HFMCD diet challenge, as indicated by liver TG and collagen

**Fig. 6 | Trim26 alleviates NASH by suppressing the Cebpd-Hif1a cascade. a** Representative western blotting bands showing the CCAAT/enhancer binding protein delta (Cebpd), hypoxia-inducible factor-1a (Hif1a), nitric oxide synthase 2 (Nos2), phosphorylated and total p38 and NF-κB p65 protein expression in livers of indicated groups (*n* = 4 per group). **b** Representative western blotting bands showing the Cebpd, Hif1a, Nos2, phosphorylated and total p38 and NF-κB p65 protein expression in livers of indicated groups (*n* = 4 per group). **c** Representative western blotting bands displaying the CEBPD, HIF1A, NOS2, phosphorylated and total P38 and P65 protein expression in indicated groups (*n* = 4 per group). **d** Representative immunoblotting bands displaying the CEBPD, HIF1A, NOS2, phosphorylated and total P38 and P65 protein expression in indicated groups (*n* = 4 per group). **e** Representative Nile red staining images displaying the lipid deposition after 10 h, 0.5 mM palmitate+1.0 mM oleic acid (PAOA)-incubated transfected or co-transfected cultured L02 cells (upper; *P* < 0.0001 by one-way ANOVA) and mice primary hepatocytes (lower; *P* < 0.0001 by one-way ANOVA) (scale bars: magnification, 100×; *n* = 10 per group). **f** Co-immunoprecipitation detection of the interaction of TRIM26 with CEBPD in L02 cells transfected with TRIM26-Flag or CEBPD-HA plasmids. Immunoblotting probed with anti-HA or anti-Flag antibody (upper). Representative immunoblotting bands for GST precipitation showing TRIM26-CEBPD binding by treating purified CEBPD-His with purified TRIM26-HA-GST or by treating TRIM26-His with purified CEBPD-HA-GST in vitro (lower). Purified GST was regarded as a control. **g** Schematic of human full-length and truncated TRIM26 and CEBPD (upper), and representative western blotting mapping assay indicating the interaction domains of TRIM26 and CEBPD (lower). **h** TRIM26 structure showing the two conservative motifs (TRIM26(CXXC), Motif-1# & TRIM26(CXXXC), Motif-2#) of RING domain in different species. **i** Representative Nile red staining images (*P* < 0.0001 by one-way ANOVA) and intracellular TG levels (*P* < 0.0001 by one-way ANOVA) showing the lipid accumulation in PAOA-treated AdGFP-, Ad*TRIM26*(AXXA)-, Ad*TRIM26*(AXXXXA)- or Ad*TRIM26*-transfected L02 cells for 10 h. The L02 cells transfected with AdGFP were used as a control (scale bars: magnification, ×100; *n* = 10 per group). **j** Representative western blotting bands displaying the TRIM26, CEBPD, HIF1A, NOS2, phosphorylated and total P38 and P65 protein expression in PAOA-treated transfected L02 cells (*n* = 4 per group). **k** qPCR assay showing the inflammation- and fatty acid synthesis-associated genes expression alteration in PAOA-treated transfected L02 cells (*n* = 4 per group; *P* < 0.05 by one-way ANOVA). Data are expressed as mean ± SEM. The relevant experiments presented in this part were performed independently at least three times. *P* < 0.05 indicates statistical significance.

content, Pearson analysis, H&E staining, and oil red O staining (*P* < 0.01 by one-way ANOVA; Fig. 8c–f). Additionally, compared to the HFMCD-fed NTG mice, HTG mice also had reduced hepatic fibrosis and decreased F4/80- and CD11b-positive inflammatory cell infiltration in response to HFMCD treatment (Fig. 8g, h). In the same way, according to previous studies and our above in vivo assay[17,22], New Zealand white rabbits were thereby subjected to in vivo-mediated gene therapy by lentivirus injection, followed by HFHC feeding for 8 weeks (Fig. 8i). As expected, rabbits with LV-*Trim26* injection via liver portal vein presented downregulated dyslipidemia, hepatic steatosis, and hepatic injury compared to those in LV-Control groups after 8 weeks of HFHC diet treatment, as confirmed by liver lipid contents detection, serum glucose and insulin assay, H&E staining, oil red O staining, and serum contents of liver function indicators (*P* < 0.01 by 2 tailed *t* test; Fig. 8j–m). These results indicated that overexpression of Trim26 is permissive for metabolism homoeostasis, and Trim26 is also critical for improving NASH progression in rodent and rabbit models.

## Discussion

Metabolic insult-associated liver disorders frequently arise as a result of excessive nutrient intake-induced obesity and type 2 diabetes (T2D). These diseases are initially recognised as NAFLD, which can significantly accelerate the progression to NASH clinical phenotypes[5,6,17]. The primary factor contributing to the failure and sluggish progress of clinical treatment medicines and related methods for NASH is our limited comprehension of the intricate aetiology of the disease. Furthermore, it is indisputable that acquiring a thorough and extensive comprehension of the molecular aetiology and advancements in research pertaining to steatohepatitis, as well as the exploration of new therapeutic targets and efficacious medications, have emerged as primary objectives in NASH treatment and have garnered significant attention from scholars worldwide[7,8]. Currently, a series of TRIM members have been highlighted by ubiquitin-related target assays in different models in vivo or in vitro. The top 4 indicators, i.e., TRIM26, TRIM31, RNF5, and TRIM8, were distinguished from the different indicated groups. Also, our previous report indicated that TRIM31 confers protection against NASH by suppressing iRhom2/MAP3K7 axis-related inflammation[23]. TRIM26 displays inhibitory effects on NASH progression by regulating CEBPD-HIF1A-related inflammation and NOS2 signalling. Although an increased inflammatory response is the major mechanism for the development of NASH, the process of endogenous inhibition of inflammation-associated signalling pathways via suppressors may be very different. Our current work expands and presents candidates for possible protective effects, providing potential experimental evidence for the treatment and drug development of NASH.

In the present work, we have successfully identified hepatocyte Trim26 as an intrinsic inhibitor of NASH diseases. The current study employed in vivo experimentation to investigate the role of liver Trim26 as a robust inhibitor of NASH pathologies induced by excessive nutrition. To ascertain the protective effects of Trim26 in rodent or rabbit NASH models, a range of experimental models were established, including hepatocyte-specific *Trim26* ablation, transgenic mice, and *Trim26*-mediated gene therapy mice. It was shown that following a prolonged challenge of a HFHC or WTDF diet, the ablation of *Trim26* in hepatocytes resulted in a significant increase in liver steatosis, hepatic inflammation, and hepatofibrosis. In contrast, hepatocytes exhibiting increased expression of Trim26 exhibited a notable deceleration in the pathological phenotypes associated with steatohepatitis. Additionally, we observed that TRIM26 interacts with CEBPD and inhibits its activation through the process of ubiquitin proteasome degradation. This interaction effectively hinders the CEBPD-HIF1A signalling pathway, as well as its downstream pathways involving HIF1A-NOS2, P38, and NF-κB P65, which are prone to over-activation under metabolic stresses.

In fact, the pathogenesis of NASH is complex and is considered to be a progressive disease accompanied by pathological symptoms including increased inflammation, insulin resistance, and impaired glucose tolerance[2,8,17]. Importantly, a single animal model did not fully simulate the pathological process of NASH. Thus, the best way to simulate NASH progression in an animal model is to use a series of different models for research at the same time. The method can minimise the effects of NASH pathogenesis caused by heterogeneity in the experiment results. Therefore, many targeted drugs that have entered clinical trials have failed to achieve satisfactory results. Lately, increasing evidence has revealed that CEBPD, a member of the CEBP family of transcription factors, indeed serves as a key proinflammatory transcription factor, promoting pro-inflammatory cytokine expression. CEBPD is capable of mediating IL-1β- or collagen-triggered PTGS2 release and TLR4 signalling activation, leading to the production of TNF-α, MIP-1, and CCL2, and macrophage inflammatory infiltration[19,20,24]. Furthermore, in keeping with its pro-inflammatory effects in vitro, CEBPD also plays a critical role in the progression of hepatic steatosis, including the regulation of liver inflammation and lipid metabolism[20,24]. The over-activated CEBPD induced by metabolism insults facilitates the deterioration of metabolic profiles, elevated insulin resistance, hepatosteatosis, and inflammatory responses[20,21,24]. Of note, previous studies also determined that metabolic stimulation-

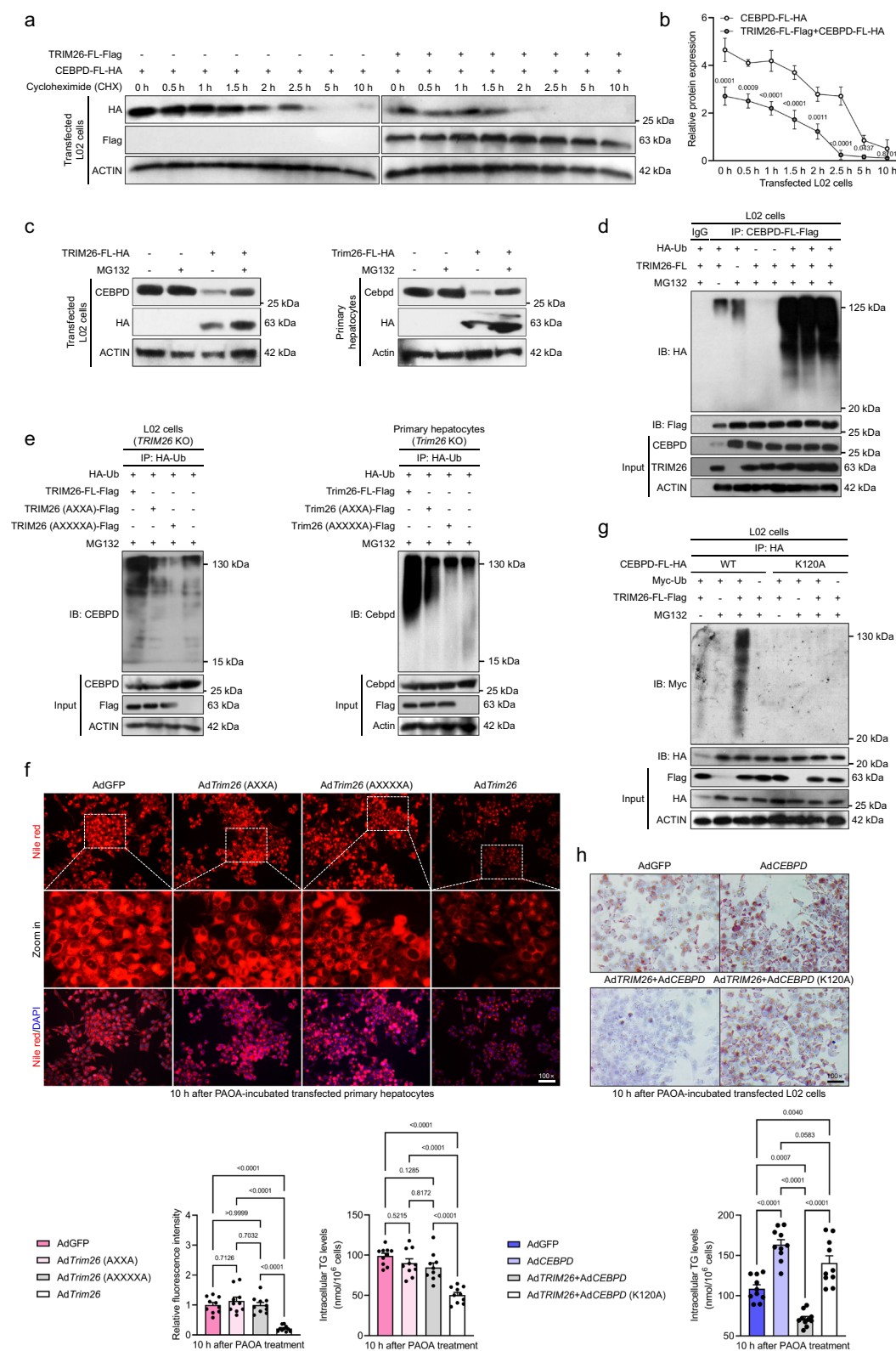

induced upregulation of CEBPD could promote HIF1A transcription and signalling activation, resulting in a boost in downstream pro-inflammatory and oxidative stress cascades[20,25]. However, the regulatory mechanism of CEBPD in the occurrence and development of NASH and the potential factors that can regulate the expression of CEBPD abundance are still unclear. Animatingly, TRIM26 specifically inhibited CEBPD signalling, which led to the alleviation of NASH progression and its complications not only in the liver but also

systemically. The hepatocyte-specific *Trim26* deletion interrupts hepatic metabolism homoeostasis, is accompanied by elevated lipid deposition, the glucose metabolic syndrome, and liver inflammation, and dramatically accelerates the development and progression of NASH. In contrast, adeno-associated virus-, adenovirus-, and transgenic overexpressed-induced *Trim26* gene therapy suppresses NASH in rodent or rabbit models. In line with metabolism stresses, mechanistically TRIM26 interacts with CEBPD, and couples

**Fig. 7 | TRIM26 interacts with CEBPD, leading to proteasome-dependent CEBPD polyubiquitination degradation. a** Human L02 cells were transfected with TRIM26-FL-Flag and CEBPD-FL-HA as indicated, and 10 h later cycloheximide (CHX) (100 μM) was incubated for the indicated time frame. **b** Related to western blotting bands in (**a**), relative protein expression levels for CEBPD-FL-HA and TRIM26-FL-Flag+CEBPD-FL-HA in transfected L02 cells after time-course treatment in vitro ($n = 4$ samples per group; $P < 0.05$ by one-way ANOVA). **c** Immunoblotting detection of TRIM26-FL-HA transfected L02 cells and Trim26-FL-HA transfected mice primary hepatocytes with/without CEBPD or Cebpd plasmids, and MG132 (25 μM, 10 h) treatment. **d** Representative immunoprecipitation and immunoblotting assays of the indicated proteins in the L02 cells transfected with TRIM26-FL, CEBPD-FL-Flag or HA-tagged ubiquitin (Ub) plasmids, and 10 h later treated with MG132. **e** Representative immunoprecipitation and immunoblotting bands showing the indicated protein in the *TRIM26*-deficient L02 cells transfected with TRIM26-FL-Flag, TRIM26 (AXXA)-Flag, TRIM26 (AXXXXA)-Flag and HA-tagged ubiquitin

vectors, and MG132 (25 μM, 10 h) treatment (left). The corresponding in vitro detection was performed in *Trim26*-deficient primary hepatocytes after indicated vectors transfection (right). **f** Representative Nile red staining images ($P < 0.0001$ by one-way ANOVA) showing the lipid accumulation ($P < 0.0001$ by one-way ANOVA) in PAOA-treated transfected primary hepatocytes (scale bars: magnification, 100×; $n = 10$ images per group). **g** L02 cells were transfected with CEBPD-FL-HA or K120A-CEBPD-HA, along with Myc-Ub and TRIM26-FL-Flag, as indicated, and treated with/without MG132. **h** Representative oil red O staining images showing the lipid accumulation in PAOA-treated transfected L02 cells (scale bars: magnification, 100×; $n = 10$ per group; $P < 0.0001$ by one-way ANOVA) and corresponding intracellular TG levels ($n = 10$ per group; $P < 0.0001$ by one-way ANOVA). Data are expressed as mean ± SEM. The relevant experiments presented in this part were performed independently at least three times. $P < 0.05$ indicates statistical significance.

polyubiquitination to promote CEBPD degradation, thus decreasing CEBPD protein abundance in hepatocytes and its downstream CEBPD-HIF1A signalling activation.

TRIM26, an important member of the E3 ubiquitin ligase family, was treated as a multi-faceted factor in many pathological or physiological processes, e.g., diabetes and multiple sclerosis, by mediating different ubiquitination modifications of potential targeted proteins, leading to signal transduction and targeted substrate degradation[10,11,13–15]. Our results uncovered that TRIM26 activity and protein expression in human subjects and rodent models with NASH pathological features were markedly reduced in the context of metabolic stress challenges. Consistently, previous reports have revealed that the elevated TAB1- or TLRs-induced upregulation of proinflammatory signalling may be attributed to TRIM26 activity[10,15,19]. Also, TRIM26 has been shown to suppress interferon-β (IFN-β) production and the antiviral response by targeting nuclear interferon regulatory factor 3 (IRF3) via polyubiquitination and degradation of nuclear IRF3[11]. Notably, in vivo overexpression of TRIM26 in livers significantly mitigates carbon tetrachloride-induced liver fibrosis progression by suppressing hepatic stellate cells (HSCs) hyperactivation via degradation of solute carrier family-7 member-11 (SLC7A11) and promotion of HSCs ferroptosis[14]. In line with these findings, our current data further demonstrated that TRIM26 exhibited anti-inflammatory actions and positive metabolic homoeostasis regulation by elevating CEBPD degradation via an increase in polyubiquitination and suppressing its downstream CEBPD-HIF1A signalling activation. Moreover, the inhibitory effects of TRIM26 on CEBPD are tightly dependent on its catalytic function. TRIM26 protein abundance and activity are negatively correlated with CEBPD expression, elevated liver inflammation, and circulating lipid contents in human subjects with NASH phenotypes, rodent NASH models, and rabbit models. In the context of in vivo investigations, it was observed that mice lacking Trim26 had heightened levels of liver injury. Although the pro-inflammatory reactions we are investigating with TRIM26 excision may potentially occur due to the inactivation of additional TRIM26 substrates linked with inflammation, such as SLC7A11 and TBK1, none of these putative targets have been shown to significantly accelerate inflammation to the same extent as CEBPD. Additionally, our study provided further evidence supporting the notion that TRIM26 functions as a very effective suppressor protein in conjunction with CEBPD. The action of TRIM26 was observed to inhibit the production of CEBPD and subsequent activation of CEBPD-HIF1A signalling pathway by promoting the degradation of CEBPD through the proteasome. Interestingly, the mutation of two conserved motifs, specifically TRIM26 (AXXA) and TRIM26 (AXXXXA), was found to have no effect on the ubiquitination of CEBPD in both in vivo and in vitro experiments. This suggests that the RING domains play a crucial role in regulating the protective function of TRIM26 on downstream CEBPD-HIF1A signalling in the context of steatohepatitis. The findings of this study provide

additional evidence that the fundamental catalytic role of two conserved motifs within the RING domain of TRIM26 could serve as a specific target for inhibiting the development of NASH and its related metabolic syndrome produced by metabolic stress.

In conclusion, our findings, as depicted in Fig. 9, indicate that TRIM26 serves as a viable regulator in the elimination of CEBPD during the pathological advancement of NASH. The liver protein TRIM26, particularly its RING domain, engages in an interaction with CEBPD and facilitates its degradation through enhanced polyubiquitination, resulting in the activation of downstream CEBPD-HIF1A signalling. This activation ultimately leads to the amelioration of steatohepatitis diseases. The existing signalling pathway presents a potentially crucial target for the treatment of NASH and offers theoretical insights for the development of pharmaceutical interventions for this globally prevalent condition.

## Methods
### Study design and ethical statement
The experimental protocols pertaining to the utilisation of animals in this study were granted approval in accordance with the guidelines outlined in the Guide for the Care and Use of Laboratory Animals (8th edition NIH, in Chinese). Additionally, the Institutional Animal Use and Care Committee (IACUC) at Chongqing University of Education (20190012CQUE), Chongqing University, Shandong Cancer Hospital and Institute, Shandong First Medical University & Shandong Academy of Medical Science, Shandong University of Traditional Chinese Medicine, and Fudan University provided permission for the aforementioned protocols. The methodologies and protocols employed in the present study were implemented in compliance with the Regulations of the People's Republic of China on the Administration of Experimental Animals (Revised & Exposure Draught), as stipulated by the Ministry of Science and Technology of the People's Republic of China (http://www.most.gov.cn).

### Human donors liver samples
Liver specimens from adult donors with non-alcoholic fatty liver disease (NAFLD) were obtained by the collection of biopsy tissue samples or liver transplantation samples. The non-steatotic liver tissues that were pertinent to the study were procured from donors who were deemed ineligible for liver transplantation due to reasons unrelated to liver conditions. A total of 16 non-steatosis samples, 17 simple steatosis samples, and 16 NASH phenotypic liver samples were procured and included in the present investigation. The human liver samples used in the present manuscript are the same as those used in our previous paper[23]. It is important to mention that liver samples exhibiting steatosis from individuals with the following conditions were excluded from the study: (i) excessive alcohol consumption (defined as consuming more than 70 g of alcohol per week for females or more than 140 g of alcohol per week for males), (ii) viral infection or drug abuse

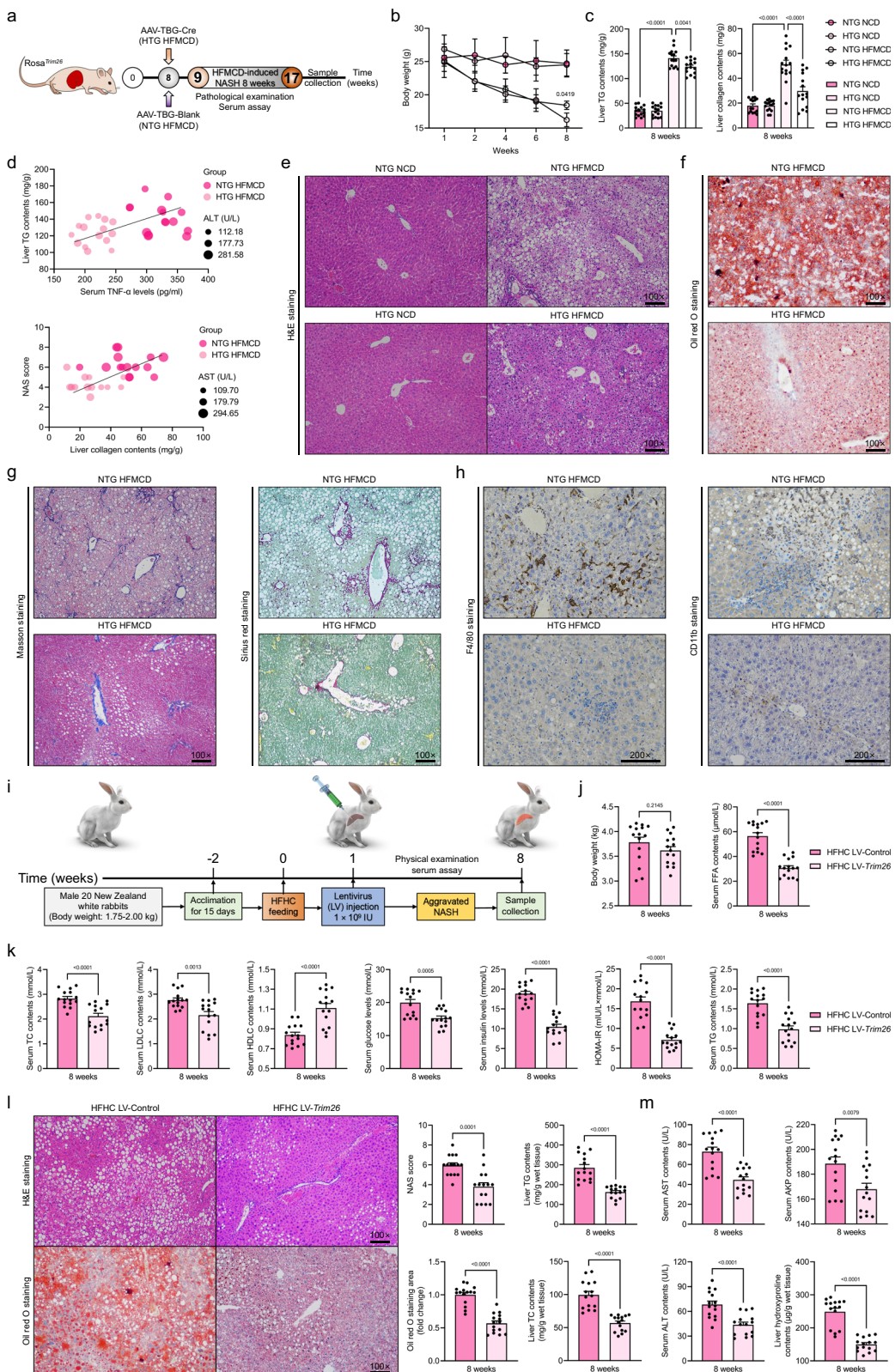

(including hepatitis B and C virus infection), long-term use of lipid-lowering, blood glucose regulation, or anti-inflammatory medications (for a duration of 24 months or longer). There are two other causes of steatosis that should be considered: autoimmune hepatitis and other factors such as primary biliary cirrhosis, uncommon metabolic diseases, etc. Additionally, liver samples were obtained from the non-steatotic portion of the livers of donors who underwent liver resection

due to the presence of liver hemangioma or liver cyst. Two pathologists independently detected hierarchical steatosis and steatohepatitis based on the grading system of standard histological criteria defined by the NASH Clinical Research Network. Patients who had NAS scores ranging from 1 to 2, together with ballooning scores of 0 and no fibrosis, were categorised as having uncomplicated steatosis. Cases exhibiting a NAS (NAFLD Activity Score) of 5 or above, or a NAS ranging

**Fig. 8 | Elevated hepatocyte Trim26 expression alleviates HFMCD diet-induced NASH phenotypes in rodent and HFHC-triggered NASH progression in rabbit model. a** Experimental design outline of AAV-TBG-Cre-induced Trim26 over-activation in livers of 8 weeks high fat plus Met & choline dual-deficient diet (HFMCD) diet (HTG HFMCD)- or normal chow diet (NCD) (HTG NCD)-fed Rosa$^{Trim26}$ mice. The AAV-TBG-Blank was used as control (NTG HFMCD; NTG NCD). **b**, **c** Records for the body weight ($P = 0.0419$ by one-way ANOVA) (**b**), liver trigly-ceride (TG) contents and hepatic collagen contents ($P < 0.0001$ by one-way ANOVA) (**c**) of the indicated mice ($n = 15$ mice per group). **d** Pearson correlation analysis showing the correlations between liver TG contents and serum TNF-α levels (upper), and NAS score and liver collagen contents (lower) in HTG HFMCD and NTG HFMCD mice. $P < 0.001$ for all of these correlations ($n = 15$ per parameter; $n = 45$ in total). Correlations were performed using Pearson rank correlation coef-ficient analysis. **e**–**g** Representative images of H&E staining (**e**), oil red O staining (**f**), masson & sirius red staining (**g**) in indicated mice (scale bars: magnification, ×100; $n = 10$ images per group). **h** Representative pictures of F4/80 and CD11b inflam-matory infiltration in HTG HFMCD and NTG HFMCD mice (scale bars: magnifica-tion, ×200; $n = 10$ images per group). **i** Experimental design outline of lentivirus (LV) injection-induced Trim26 overexpression in livers of 8 weeks HFHC diet-fed rabbits (HFHC LV-*Trim26*). The LV-loaded empty vectors were used as control (HFHC LV-

Control). **j**, **k** Records for the body weight, serum free-fatty acid (FFA) contents ($P < 0.0001$ by 2 tailed $t$ test) (**j**), and serum total cholesterol (TC) levels ($P < 0.0001$ by 2 tailed $t$ test), low density lipoprotein cholesterol (LDLC) levels ($P = 0.0013$ by 2 tailed $t$ test), high density lipoprotein cholesterol (HDLC) levels ($P < 0.0001$ by 2 tailed $t$ test), glucose levels ($P = 0.0005$ by 2 tailed $t$ test), insulin levels ($P < 0.0001$ by 2 tailed $t$ test), homoeostasis model assessment of insulin resistance (HOMA-IR) levels ($P < 0.0001$ by 2 tailed $t$ test) and TG levels ($P < 0.0001$ by 2 tailed $t$ test) (**k**) in HFHC LV-Control and HFHC LV-*Trim26* groups ($n = 15$ rabbits per group). **l** Representative images of H&E staining and oil red O staining ($P < 0.0001$ by 2 tailed $t$ test), and corresponding NAS score ($P = 0.0001$ by 2 tailed $t$ test), and liver TG ($P < 0.0001$ by 2 tailed $t$ test) & TC ($P < 0.0001$ by 2 tailed $t$ test) contents in indicated groups (scale bars: magnification, 100×; $n = 15$ images per group). **m** Records for serum aspartate aminotransferase (AST) ($P < 0.0001$ by 2 tailed $t$ test), alanine aminotransferase (ALT) ($P < 0.0001$ by 2 tailed $t$ test) and alkline phosphatase (AKP) ($P = 0.0079$ by 2 tailed $t$ test) contents, and liver hydroxyproline ($P < 0.0001$ by 2 tailed $t$ test) contents in HFHC LV-Control and HFHC LV-*Trim26* groups ($n = 15$ rabbits per group). Data are expressed as mean ± SEM. The relevant experiments presented in this part were performed independently at least three times. $P < 0.05$ indicates statistical significance.

from 3 to 4 accompanied with fibrosis, were categorised as Non-Alcoholic Steatohepatitis (NASH). Instances having a NAS value of 0 were categorised as normal. Furthermore, it should be noted that the specimens used in this investigation to represent non-steatosis and simple steatosis were obtained from individuals who had not been either statins or insulin before to sample collection. The liver samples in this cohort were obtained from individuals who exhibited the NASH phenotype. These patients had been administered pioglitazone at a dosage of 15-30 mg per day for a maximum duration of 24 months. The individuals providing liver samples, as well as their respective families, provide their consent in written form after being fully informed. The physiological parameters of the patients and the serology associated to hepatic damage are presented in Supplementary Table. 1. The pro-tocols employed in this study pertaining to human donors were based on the Ethical Principles for Medical Research Involving Human Sub-jects, specifically the Declaration of Helsinki (64th WMA general assembly). These protocols were also subjected to approval by the Academic Committee of Experimental Animal Ethics, Use & Care Union in Chongqing University of Education, Shandong Cancer Hospital and Institute, Shandong First Medical University & Shandong Academy of Medical Science, Shandong University of Traditional Chinese Medi-cine, and Fudan University.

## Reagents
The main antibodies employed in this work, targeting the indicated proteins, were procured from Abcam Inc.: anti-Actin (#ab179467) at a dilution of 1/2500, anti-Gapdh (#ab9485) at a dilution of 1/1500, anti-CEBP delta (#ab245214) at a dilution of 1/1000, anti-HIF-1 alpha (#ab179483) at a dilution of 1/1000, anti-iNOS (#ab178945) at a dilution of 1/1000, anti-p38 (#ab182453) at a dilution of 1/1000, anti-phosphorylated p38 (#ab4822) at a dilution of 1/1000, anti-NF-κB p65 (#ab32536) at a dilution of 1/1000, anti-phosphorylated NF-κB p65 (#ab76302) at a dilution of 1/1000, anti-alpha skeletal muscle actin (#ab184705) at a dilution of 1/1000, anti-CD11b (#ab133357) at a dilu-tion of 1/100, and anti-F4/80 (#ab16911) at a dilution of 1/100. Anti-bodies against TRIM26 (#PA5-62191, 1/150-1/1000 dilution) and TRIM26 (#ABIN7076014, 1/150-1/1000 dilution) were obtained from Thermo Fisher Scientific and Antibodies-online GmbH Inc., respec-tively. Moreover, the antibodies against anti-Flag (CST, #14793, 1/1000 dilution), anti-HA (Abcam, #ab9110, 1/1000 dilution), anti-Ub (Abcam, #ab134953, 1/1000 dilution) and anti-Myc (Abcam, #ab9106, 1/1000 dilution) were also used in the current study. The QuantiPro™ BCA protein quantification kits (#71285-3, Millipore®) were employed to ascertain the protein concentration of the samples. In the western

blotting analysis, we employed HRP-tagged secondary antibodies obtained from Thermo Fisher Scientific. These antibodies were diluted at a ratio of 1/10,000 to 1/15,000 for the purpose of visualisation. The palmitate (PA) and oleic acid (OA) were acquired from Sigma-Aldrich. The TaqMan® Universal PCR Master Mix (#P/N 4304437) and Pow-erUp™ SYBR™ Green (#A25742) were procured from Applied Biosystems.

## Mouse strains
So as to generate viable mice with a targeted deletion of *Trim26*, the *Trim26$^{flox/flox}$* (Flox) mice were developed using the CRISPR-Cas9 genetic engineering editing system, employing the C57BL/6 strain as the genetic background. Conditional knockout experiments were conducted by specifically targeting exons 3-5 of the *Trim26* gene for conditional deletion. In summary, the specific exons of *Trim26* were surrounded by two *loxP* sites, resulting in the identification of two distinct single guide RNAs, namely guide RNA1 and guide RNA2, which were designed to target the introns of Trim26. The plasmids that were packaged had *Trim26* exons flanked by two *loxP* sites, together with two homology arms. These plasmids were considered as the matching template. The zygotes were subjected to coinjection of the targeting vector, guide RNA1, guide RNA2, and Cas9 in order to establish con-ditional deletion mice. The resulting offspring, which have specific exons surrounded by two *loxP* sites on one allele, were used to gen-erate *Trim26$^{flox/flox}$* mice. The generation of hepatocyte-specific *Trim26* ablation (HKO) offspring was achieved through the mating of *Trim26$^{flox/flox}$* offspring with albumin-cre tool transgenic mice (Alb-Cre) obtained from Cyagen Biosciences in Guangzhou, China. Littermates with the *Trim26$^{flox/flox}$* genotype were selected as appropriate control subjects for the acquired HKO mice, serving as a basis for comparison.

With the objective of generating mice with conditional over-expression of Trim26, the Rosa$^{Trim26}$ mice were developed on the C57BL/6 strain. This was achieved by performing a *Trim26* conditional knockin at the *Rosaβgeo26* locus in mice using the CRISPR-Cas9 genetic engi-neering editing technique. In summary, the cassette comprising *Rosaβgeo26*-(pCAG)-*loxp*-STOP-*loxp*-m*Trim26*-pA box was introduced into the first intron of *Rosaβgeo26*. Following that, the targeting vector, guide RNA, and Cas9 were co-injected into spermatogonia to generate the Rosa$^{Trim26}$ mouse. The experiments involved inducing the condi-tional overexpression of Trim26 in hepatocytes (HTG) through the intravenous injection of an adeno-associated virus serotype-8 (AAV8)-thyroxine-binding globulin promoter (TBG)-recombinase Cre vector (AAV8-TBG-Cre). The level of overexpression was subsequently deter-mined using an immunoblotting assay. The equivalent control group for

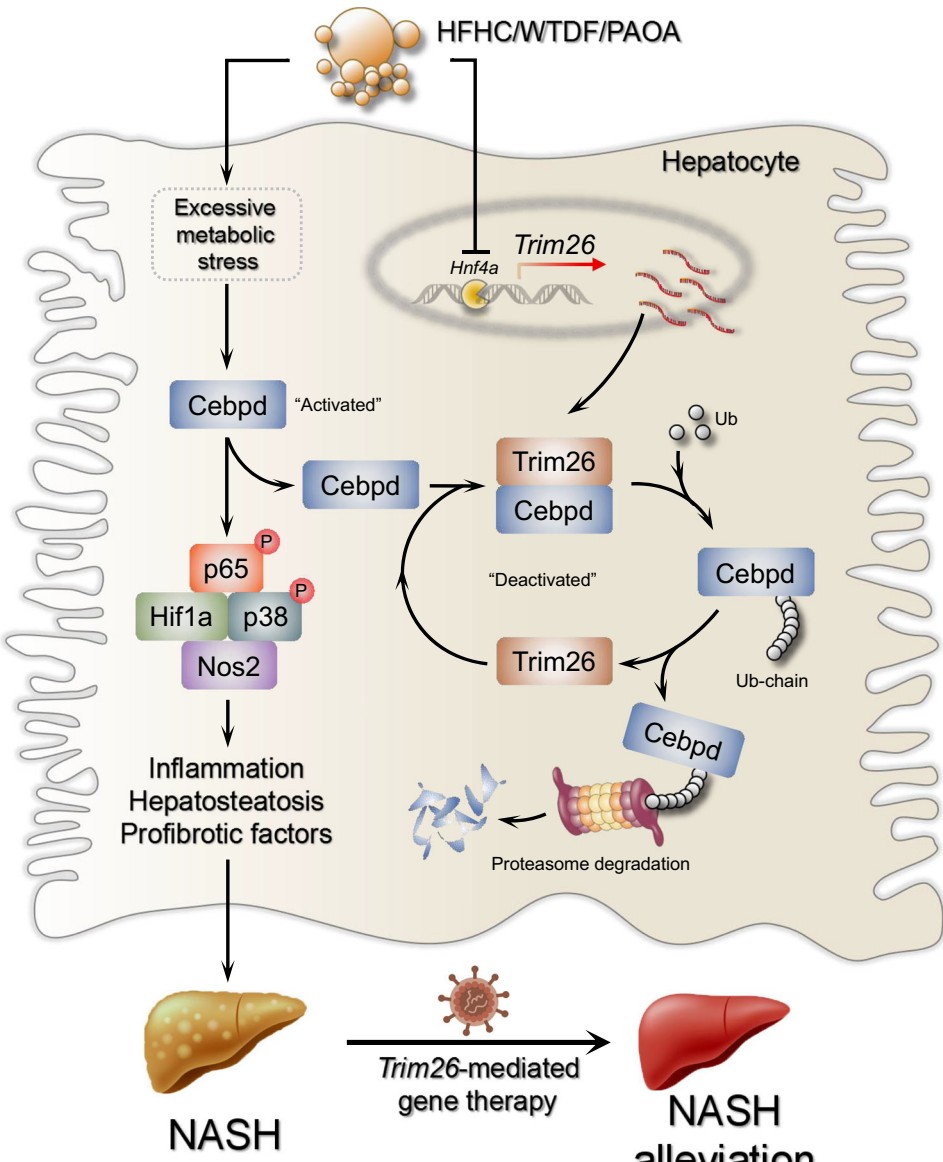

**Fig. 9 | Schematic diagram showing the protective function of Trim26 in NASH progression.** Excessive metabolic insults, e.g., western-type diet+15% fructose drinking water (w/v) (WTDF), high fat+high cholesterol diet (HFHC), and 0.5 mM palmitate+1.0 mM oleic acid (PAOA) markedly upregulate Cebpd abundance, subsequently leading to downstream cascade pathway including Hif1a/NF-κB p65/p38 and Nos2 activation. Over-activated Hif1a/NF-κB p65/p38 and Nos2 signalling contributing to increase of lipid deposition, profibrosis- and proinflammation-associated cytokines and chemokines. Thus the proinflammatory and profibrotic factors, e.g., IL-6, IL-1β, TGFβ and TNF-α produced from impaired hepatocytes promote hepatofibrosis development. Conversely, forced increase of Trim26 in hepatocytes significantly block and directly interact with Cebpd to facilitate proteasome-mediated protein degradation, thereafter suppressing Cebpd abundance and its downstream cascade pathway activation.

the obtained HTG mice consisted of littermates that received AAV-blank injection, namely Rosa$^{Trim26}$ littermates.

With the goal of conducting feasible experiments with mice with a conditional deletion of *Cebpd* (HKO), *Cebpd*$^{flox/flox}$ animals on the C57BL/6 strain were generated using the CRISPR-Cas9 genetic engineering editing technology. The first exon of the Cebpd gene was selected as the specific location for conditional deletion to carry out a conditional knockout experiment. In conclusion, exon 1 is recognised, wherein exon 1 contains the ATG start codon and the TAA termination codon. Two *loxP* loci were positioned on either side of the targeted *Cebpd* exon, resulting in the identification of two single guide RNAs, namely guide RNA1 and guide RNA2, which were designed to target *Cebpd* exon 1. The generation of the targeted vector will involve the utilisation of PCR to engineer the homologous arms and cKO region, with the BAC clone serving as the template. The targeting plasmids, guide RNA1 and guide RNA2, were co-injected with Cas9 into zygotes to establish conditional deficient animals. The resulting offspring, which possessed a specific genetic modification including the insertion of *loxP* sites flanking exon 1 on one allele, were used to generate mice with a *Cebpd*$^{flox/flox}$ genotype. Hepatocyte-specific *Cebpd* ablation (HKO) mice were created through the breeding of *Cebpd*$^{flox/flox}$ mice with transgenic mice expressing the albumin-cre tool (Alb-Cre) (Cyagen Biosciences, Suzhou, China). The *Cebpd*$^{flox/flox}$ (Flox) mice from the same litter were used as control subjects to compare with the acquired HKO mice. The dual-deficiency (DHKO) mice with hepatocyte-specific *Trim26* and *Cebpd* deficiencies were generated through the breeding of *Cebpd*$^{flox/flox}$ mice with HKO (*Trim26*) mice. The progeny lacking expression of the Trim26 and Cebpd proteins were detected and chosen through the implementation of a western blotting assay. These selected progeny were subsequently utilised in further investigations

as stated. In addition, the mice utilised in this study were of the wild-type C57BL/6 strain, namely male mice aged 6-8 weeks. These mice were procured from Beijing Vital River Laboratory Animal Technology Co., Ltd., located in Beijing, China.

### Establishment of nonalcoholic steatohepatitis (NASH) animal model

The current study utilised three distinct rodent models of non-alcoholic steatohepatitis (NASH) and one model of NASH in rabbit to conduct the respective investigations.

### High fat plus high cholesterol diet (HFHC)-mediated NASH model

A murine model exhibiting NASH phenotype was established by subjecting the mice to a HFHC diet consisting of 42% saturated fat, 14% protein, 44% carbs, and 0.2% cholesterol by weight for a duration of 16 weeks[17,23]. The mice used in the experiment were administered a conventional normal chow diet (NCD) from Research Diets, USA (Cat: D12450J) for a continuous period of 16 weeks. This was done in order to establish a control group that would correspond to the obtained HFHC-fed model. In the in vivo experiment involving rabbits, a modified version of a previous study was followed[26–28]. Male New Zealand white rabbits were administered a specific HFHC diet for a duration of 8 weeks. This HFHC diet consisted of a standard diet supplemented with 2% maltodextrin, 2% cholesterol, and 10% saturated fats (Cat: 621079; Dyets, Bethlehem, Pa). The purpose of this dietary intervention was to establish a rabbit model of steatohepatitis.

### Western-type diet plus 15% fructose (w/v)-drinking water (WTDF)-induced NASH model

The second mice model exhibiting NASH phenotype was established by administering a diet consisting of WTDF (Cat: D12079B; Research Diets, Inc., New Brunswick, USA) supplemented with a 15% w/v fructose solution (Cat: 630030391, Sinopharm Chemical Reagent Co., Ltd., China) in the form of drinking water (Cat: 1010004000, C'estbon, China) for a duration of 16 weeks[17,23]. The control group of mice was provided with a standard chow diet.

### High fat plus Met & choline dual-deficient diet (HFMCD)-induced NASH model

The third mouse model exhibiting NASH phenotype was established through an 8-week dietary intervention using a HFMCD diet (HFMCD, Cat: A06071301B, Research Diets)[17,23]. The control group of mice was provided with a chow diet that was specific to their nutritional needs (referred to as NCD, or speciality feeds).

### Animals treatment and model design

To attempt to mitigate the impact of hormonal fluctuations on metabolic processes, exclusively male animals within the age range of 6-8 weeks were utilised for all experimental procedures conducted in this study. Prior to the commencement of the experiment, the animals involved in the respective study were subjected to a period of acclimation to their housing conditions lasting for a duration of 7 days. The animals were housed in controlled conditions to maintain a consistent temperature and moisture level. The temperature was regulated using a Haier central air conditioning system (Cat: RFC140MXSCVD/G, China), while the environment was kept aseptic. The cages were maintained at a temperature of 25 °C and a humidity level of 55%-60%. The animals were subjected to a regular light/dark cycle of 12 h light and 12 h dark. They had access to unlimited pathogen-free drinking water (Cat: 1010004000, C'estbon, China) and were provided with standard fodder in their enclosures.

**Animal experiment model 1.** Male mice of the wild type (WT) were selected, aged between 6 to 8 weeks. These mice were subjected to a HFHC or WTDF diet for a duration of 16 weeks. The purpose of this dietary intervention was to induce a non-alcoholic steatohepatitis (NASH) phenotype in the mice. The remaining wild-type (WT) mice were administered a NCD diet for a duration of 16 weeks to serve as control subjects for comparison purposes. Following the completion of the experiment, the liver tissue was collected from the designated mouse model in order to analyse the associated signal indications.

**Animal experiment model 2.** To continue examining the potential protective effects of Trim26 against steatohepatitis pathologies induced by a HFHC diet, we conducted ex vivo-mediated administration of *Trim26* gene therapeutics using lentivirus-packaged full-length *Trim26* cDNA sequences (LV-*Trim26*) or short hairpin RNA (shRNA) targeting *Trim26* (LV-sh*Trim26*). These therapeutics were transplanted into WT mice that had been pretreated with an 8-week HFHC diet The ex vivo-mediated gene therapies approaches described in our previous publication were implemented accordingly[17,23]. Subsequently, the experimental mice underwent a continuous fasting period of 8 h, followed by an examination of their blood glucose and insulin levels.

**Animal experiment model 3.** The *Trim26* deletion (HKO) mice were subjected to a 16-week treatment with a HFHC or WTDF diet to establish a steatohepatitis phenotype. To acquire the conditional *Trim26* gain-of-function animals, the mice were fed either a HFHC diet or a WTDF diet. The Rosa$^{Trim26}$ animals were administered a dosage of AAV8-TBG-Cre vectors (#SL101528, SignaGen Laboratories) via the tail vein, resulting in the injection of $1.5 \times 10^{12}$ genome copies (gc). This intervention aimed to induce hepatocytes-specific overexpression of *Trim26* (HTG HFHC, HTG WTDF). The control group consisted of Rosa$^{Trim26}$ mice that were administered an equivalent dosage of AAV empty vector.

**Animal experiment model 4.** The additional animal research was achieved by introducing either the full-length mouse *Trim26* sequences or mouse *Trim26* sequences with a deleted UBA domain into an AAV8 vector, resulting in the creation of AAV-TBG-*Trim26* or AAV-TBG-*Trim26* (△RING) vectors, respectively. Subsequently, the HKO$^{Trim26}$ mice were administered an injection with a dose of matching vectors equivalent to $1.5 \times 10^{12}$ genome copies (gc), resulting in the generation of *Trim26* gain-of-function mice. The mice were subsequently administered a 16-week WTDF diet content to establish the steatohepatitis phenotype. This was done for both the gain-of-function (GOF) WTDF group and the delta gain-of-function (△GOF) WTDF group. The control group consisted of mice that were injected with blank vectors.

**Animal experiment model 5.** The male mice with the DHKO (*Trim26-Cebpd*) genotype, HKO (*Trim26*) genotype, HKO (*Cebpd*) genotype, and Flox genotype were subjected to a 16-week HFHC diet. This dietary intervention aimed to induce a NASH phenotype in the mice and assess the resulting physiological and pathological alterations. Additionally, the control mice in this study were the littermates that received NCD fodder for the same duration.

**Animal experiment model 6.** The Rosa$^{Trim26}$ mice were subjected to a HFHC or HFMCD diet for a duration of 8 weeks to induce the another steatohepatitis model with varying characteristics. The Rosa$^{Trim26}$ murine models were administered with a dosage of $1.5 \times 10^{12}$ genome copies (gc) of AAV8-TBG-Cre vectors into the caudal vein in order to generate a specific overexpression of Trim26 in hepatocytes (HTG HFHC or HTG HFMCD). The control group consisted of Rosa$^{Trim26}$ mice that were administered an equivalent dosage of AAV empty vector, and were subjected to either NTG HFHC or NTG HFMCD treatment.

**Animal experiment model 7.** In the in vivo experiment involving rabbits, the methodology was based on a previous study with some

adjustments[26–28]. A total of 20 male New Zealand white rabbits weighing between 1.75 and 2.00 kg were used. These rabbits were subjected to a specific HFHC diet for a duration of 8 weeks. The HFHC diet consisted of a standard diet supplemented with 2% maltodextrin, 2% cholesterol, and 10% saturated fats (Cat: 621079; Dyets, Bethlehem, Pa). The purpose of this dietary intervention was to establish a rabbit model of steatohepatitis.

All the experimental mice were sacrificed by exsanguination under sodium pentobarbital anaesthesia (20 mg/kg *i.p.*) at the end of the experimental period.

## Cell Culture and Administration
The L02 cell line was acquired and employed in the present investigation, as previously documented in our published studies[17,23]. The L02 cell line, which is a human normal hepatocyte cell line, was acquired from the Type Culture Collection of the Chinese Academy of Sciences in Shanghai, China. The cell lines utilised in our laboratory were subjected to a maximum of 30 passages after resuscitation. The cells that were used in both in vitro and in vivo tests were systematically assessed for potential contamination by mycoplasma using polymerase chain reaction (PCR) detection. The L02 cells were cultured in Dulbecco's Modified Eagle Medium (DMEM) supplemented with 1% penicillin and streptomycin, 10% foetal bovine serum (FBS), and maintained in a 5% $CO_2$, 37 °C directly-heated type cell incubator. The DMEM medium used was obtained from Bio-Channel Biotechnology Co., Ltd., China (Cat: BC-M-005), the penicillin and streptomycin were obtained from Biosharp Life Sciences (Cat: BL505A), and the FBS was obtained from WISENT (Cat: 085-150). The cell incubator used was manufactured by SANYO. The mouse primary cultured hepatocytes utilised in this study were obtained and concentrated from animals involved in the specified investigations using liver perfusion. In summary, following the administration of sodium pentobarbital anaesthesia, the abdominal cavity of the mice was surgically accessed. Subsequently, the liver samples were perfused at a slow rate using a 1× liver perfusion working solution (Cat: 17701038, Gibco™) and a 1× liver digest working solution (Cat: 17703034, Gibco™) through the portal vein. Subsequently, the liver tissue that had undergone digestion was subjected to filtration using a steel mesh with a pore size of 100 μm. The primary isolated hepatocytes were obtained by collecting the filtrate solution following centrifugation at 800 *g* for 5-10 minutes at a temperature of 4 °C. Subsequently, the hepatocytes were purified using a percoll-solution (Cat: 40501ES60, YEASEN, Shanghai, China). The extracted hepatocytes were grown in Dulbecco's Modified Eagle Medium (DMEM) supplemented with 10% FBS and 1% penicillin-streptomycin. The cells were maintained in a controlled environment at a temperature of 37 °C and a carbon dioxide ($CO_2$) concentration of 5%. To replicate the hepatic lipid deposition and steatosis observed in vivo, the L02 cells and mouse primary isolated hepatocytes were treated with a specific dose of PAOA (a mixture of 0.5 mM palmitate and 1.0 mM oleic acid), TNF-α, or fructose. This treatment aimed to examine the expression of the Trim26 protein and the accumulation of lipids in cells transfected with Ad*Trim26* or Adsh*Trim26*.

## Construction of the knockout cell lines
The targeted gene-deletion cell lines utilised in this study were established according to previously published methods[17,23]. In this study, cell lines were generated using the CRISPR-Cas9 technology to ablate *TRIM26*. The small guide RNA (sgRNA) targeting the human *TRIM26* gene was created and subsequently inserted into lentiCRISPR v2 vectors obtained from Addgene, located in Watertown, MA, USA. This process was carried out to generate the Cas9/sgRNA-loaded lentivirus. The design of single guide RNAs (sgRNAs) for the purpose of targeting the *TRIM26* gene was accomplished through the utilisation of the

CRISPR Design Tool, which is accessible via the following web address: http://crispr.mit.edu/. The targeted sequence zone for TRIM26 was chosen to be Exon 1. The oligonucleotide DNAs used for the sgRNAs were inserted into lentiCRISPR v2 vectors that had been subjected to digestion by the BsmBI restriction enzyme.The clones that were acquired were identified using sequencing, utilising a primer with the sequence 5′-GGACTATCATATGCTTACCG-3′, which corresponds to the U6 promoter sequence responsible for driving the expression of sgRNAs. The cell clones harbouring gene deficiencies were identified by the utilisation of western blotting. Mouse primary hepatocytes with *Trim26* deletion were obtained directly from mice that had a global deficiency of *Trim26*.

## Vectors preparation and transfection
With the goal to achieve an increased expression of *TRIM26*, the full-length *TRIM26* cDNA expression vectors of *Homo sapiens* were generated by PCR-based cDNA amplification. These vectors were subsequently inserted into the pcDNA3.1 plasmids, which were tagged with 3×HA or 3×Flag (obtained from Addgene, Watertown, MA, USA). Abridged TRIM26 and CEBPD tiny fragments expression plasmids including TRIM26-FL-HA, CEBPD-FL-Flag, TRIM26-FL-Flag, CEBPD-FL-HA, CEBPD-FL(K120A)-HA, TRIM26 (AXXA)-FL-Flag, TRIM26 (AXXXXA)-FL-Flag, CEBPD-△TAD-Flag, CEBPD-△BLZ-Flag, CEBPD-△LZR-Flag, TRIM26-△RBB-HA, TRIM26-△BB-HA and TRIM26-△CPS-HA were accordingly prepared (Fig. 6g). The plasmids for expressing WT ubiquitin with HA-tag or Myc-tag were constructed based on the pcDNA3.1 plasmid. Subsequently, the ubiquitin protein was introduced into the pcDNA3.1 Myc-tag vector obtained from Addgene, located in Watertown, MA, USA. The transfection of vectors into L02 cells was performed using ViaFect™ Transfection Reagent (Cat: E4982, Promega, Beijing, China) in accordance with the manufacturer's instructions. Furthermore, to conduct a more comprehensive investigation into the biological impacts of TRIM26 in an artificial laboratory setting, we have successfully developed specific adenovirus (Ad)-encapsulated gene expression plasmids. *Homo sapiens* full-length *TRIM26* sequences and designed short hairpin RNA (shRNA) targeting human *TRIM26* (sh*TRIM26*) (shRNA sequences RNA-1: GATGGA-TATGACGACTGGGAA; RNA-2: GATGGATATGACGACTGGGAA), adenovirus expressing shRNA for silencing of *Mus musculus Trim26* (Cat. No: shADV-274978, Vector Biolabs, Malvern, PA), human WT *TRIM26* sequences with AXXA and AXXXXA motifs of RING domain mutation, *Homo sapiens* full-length *CEBPD* sequences, and adenovirus expressing shRNA for silencing of *Mus musculus Cebpd* (Cat. No: shADV-255245, Vector Biolabs, Malvern, PA) were respectively packaged into adenovirus by Adeno-X™ Adenoviral System 3 Kit (Cat: 632269, Takara Bio Inc.). The control groups for expression suppression (knockdown) or overexpression were represented by the empty adenovirus constructs, AdshRNA and AdGFP, respectively. The adenovirus (Ad) vectors that were produced were purified and quantified to a titre of 5×10[10] plaque-forming units (PFU) using the Vivapure AdenoPACK purification kit (Cat: VS-AVPQ022, Sartorius, Shanghai, China) in accordance with the manufacturer's specifications. To create the LV-*Trim26* or LV-sh*Trim26* vectors, the cDNA sequences of mouse *Trim26* or shRNA sequences targeting mouse *Trim26* were inserted into the pLenti-GFP-Puro-CMV or pLenti-U6-GFP vectors (obtained from Addgene). These vectors were utilised to modulate the expression of Trim26 in in vivo investigations.

## Metabolic factors and serum cytokines parameters
The quantification of triglyceride (TG) contents was performed using either the Triglyceride (TG) Content Detection Kit (Cat: D799795-0050, Sangon Biotech, Co., Ltd., Shanghai, China) or the Triglyceride (TG) Assay Quantification Kit (Cat: KA0847, Novus Biologicals), in accordance with the specifications provided by the respective

manufacturers. The blood glucose levels following the trial period were analysed using universal glucose test strips (utilising the glucose oxidase method) with the specific lot number 3558469, from the ONETOUCH®Ultra® brand manufactured by LifeScan, Inc., located in CA, USA. The quantification of the Homoeostasis Model Assessment for Insulin Resistance (HOMA-IR) index was performed using measurements of fasting insulin and fasting blood glucose. The HOMA-IR index was determined and computed using the following formula: HOMA-IR = fasting insulin multiplied by fasting blood glucose, divided by 22.5. The GTT detection techniques were conducted in accordance with previously published studies[17,23]. The assessment of mouse chemokines and cytokines was conducted via commercially-available ELISA kits specifically designed for this purpose. The TNF-α (Catalogue Number: ab100747), IL-1β (Catalogue Number: ab197742), IL-6 (Catalogue Number: ab222503), CCL2 (Catalogue Number: ab208979), IL-18 (Catalogue Number: ab216165), and insulin (Catalogue Number: ab285341) enzyme-linked immunosorbent assay (ELISA) kits were acquired from Abcam. The quantification of serum or tissue glutamic pyruvic transaminase (GPT/ALT) (Cat: C009-2-1), glutamic oxalacetic transaminase (GOT/AST) (Cat: C010-1-1), alkline phosphatase (ALP/AKP) (Cat: A059-1-1), total cholesterol (TC) (Cat: A111-1-1), hepatic triglyceride (TG) (Cat: A110-1-1), γ-Glutamyl Transferase (GGT) (Cat: C017-1-1) and non-esterified fatty acids (NEFA) (Cat: A042-2-1) was performed using commercially-available kits (Nanjing Jiancheng Bioengineering Institute, China) according to the manufacturer's instructions in the specified experimental groups. The detection of liver hydroxyproline and collagen was conducted using the Hydroxyproline Assay Kit (Cat: MAK008, Sigma-Aldrich), Collagen Assay Kit (Cat: MAK322, Sigma-Aldrich), and Soluble Collagen Quantification Assay Kit (Cat: CS0006, Sigma-Aldrich), following the instructions provided by the respective products.

## Histopathological analysis

The tissue was fixed using a 4% formaldehyde-tissue fixative solution (Cat: 80096618, Sinopharm Chemical Reagent Co., Ltd., China), followed by embedding in paraffin wax (Cat: 69018961, Sinopharm Chemical Reagent Co., Ltd., China). Subsequently, transverse sections were made for histopathologic and immunohistochemical analysis. The tissue samples underwent H&E staining using the Hematoxylin-Eosin/HE Staining Kit (Cat: G1120, Solarbio Life Sciences, Beijing, China) in order to observe and assess the extent of lipid buildup and hepatic inflammation. In order to conduct a more comprehensive examination of hepatic lipid deposition, the liver slices were preserved by freezing them in tissue opti-mum cutting temperature (O.C.T)-freeze medium (Cat: C1400, Applygen Technologies, Inc., China). Subsequently, the frozen slices were treated with the oil red O Kit (Cat: PH1226, Scientific Phygene®, China) for a duration of 10-15 minutes. Following the application of a 60% isopropanol solution (Cat: 40064360, Sinopharm Chemical Reagent Co., Ltd., China) for washing purposes, the tissue slices underwent restaining using haematoxylin. In addition, to detect the collagen contents in liver tissue, the slices were treated with Masson's staining (Cat: abs9348, Masson's Trichrome Stain Kit, Absin, Shanghai, China) and Sirius red staining (Cat: PH1099, Enhanced Sirius Red Staining Kit, Scientific Phygene®, China). In order to conduct immunohistochemical analysis, the paraffin-embedded slices were subjected to dewaxing prior to being incubated with primary antibodies. The primary antibodies used included anti-TRIM26 (Thermo Fisher Scientific, #PA5-62191, diluted at a ratio of 1/150), anti-TRIM26 (Antibodies-online, #ABIN7076014, diluted at a ratio of 1/150), anti-HNF4A (Abcam, #ab201460, diluted at a ratio of 1/150-1/250), anti-F4/80 (Abcam, #ab16911, diluted at a ratio of 1/150-1/200), and anti-CD11b (Abcam, #ab133357, diluted at a ratio of 1/150-1/200). The incubation with these primary antibodies took place overnight at a temperature of 4 °C. The secondary antibody utilised in this study was either anti-

rabbit IgG or anti-mouse IgG antibodies, obtained from Thermo Fisher Scientific. The histological tests were conducted in accordance with established protocols outlined in the reagent instructions and operation manual. These studies were carried out by three histologists who were unaware of the treatment techniques. The images were observed using a light microscope (Leica, Germany) to assay sample sections, and an immunofluorescence microscope system (Leica, Germany) to detect immunofluorescence sections.

## Glutathione S-transferase (GST) pull-down detection

The interaction between TRIM26 and CEBPD was investigated through the implementation of GST pull-down analysis. In this investigation, the MagneGST™ Pull-Down System Kit (#V8870, Promega, Beijing, China) was employed to investigate protein binding. In this section, the *Rosetta*-DE3 competent cells were subjected to transformation with either the plasmid pGEx-4T-1/GST-TRIM26 or pGEx-4T-1/GST-CEBPD. Subsequently, the expression of the vectors was stimulated by treatment with 0.5 mM isopropyl β-D-thiogalactoside (IPTG) obtained from MedChemExpress (Cat: HY-15921) located in Shanghai, China. Next, the extractions were subjected to co-treatment with the respective GST particles for a duration of 1 h at a temperature of 4 degrees Celsius. The GST particles were further coincubated with Flag-tagged TRIM26 or Flag-tagged CEBPD proteins, which were obtained using immunoprecipitation (IP), for an additional 5 h. The elution of interacting proteins was performed using elution buffer, and subsequent western blotting analysis was conducted using an anti-Flag antibody. The control in this study consisted of *Rosetta*-DE3 competent cells expressing only the GST-tag.

## In vivo and in vitro binding ubiquitination detection

The current tests employed in vivo and in vitro binding ubiquitination detection methods, following the protocols and methodologies previously established by our research group[17,23]. The process of ubiquitination was examined using the VIVAlink™ Ubiquitin Kit (Cat: VB2952-50, Viva Bioscience, Exeter, UK) in accordance with the techniques and protocols outlined in the product manual.

## RNA extraction, quality control and quantitative PCR (qPCR) assay

The RNA from the specified liver tissue or cells was isolated using TRNzol Universal reagent (Cat: DP424, TIANGEN®, Beijing, China) following the suggested protocols provided by the product specification. The RNA samples were stored at a temperature of −80 °C for a maximum duration of 14 days. The confirmation of RNA content absorption at a wavelength of 260 nm was conducted through examination utilising a Nanodrop photometer manufactured by Tecan. The assurance of optimal RNA purity was achieved through the verification of the 260 nm/280 nm absorbance ratio, with values exceeding 2.00. Subsequently, a quantity of 1 μg of isolated RNA was subjected to reverse transcription using the universal RT-PCR Kit (M-MLV) (#RP1100, Solarbio Life Sciences, Beijing, China). The operation of inverse transcription was conducted at a temperature of 42 °C for a duration of 1 h, after which the enzyme was deactivated at a temperature of 70 °C for a period of 10 minutes. The PCR procedure was conducted with the PowerTrack™ SYBR Green Master Mix (manufactured by Applied Biosystems™) on the ABI PRISM 7900HT equipment (manufactured by Applied Biosystems). The primer sequences for important genes related to lipid metabolism, inflammation, and fibrosis were generated by Sangon Biotech, located in Shanghai, China. The fold difference values were calculated using the 2(-ΔΔCt) formula, where ΔCt indicates the disparity in cycle thresholds between the reference gene GAPDH and the gene of interest, and ΔΔCt reflects the relative change in the differences between the experimental groups and the control groups. The primer sequences were provided in Supplementary table 2. The supplementary table 3 contained the small

guide RNA (sgRNA) utilised for the human TRIM26 gene in the creation of knockout cells.

## Immunoblotting detection

With the goal to conduct immunoblotting analyses, liver tissue or cells were treated with RIPA (radio immunoprecipitation assay) lysis buffer (Catalogue Number: PH0317, Scientific Phygene®) to obtain lysates. Subsequently, the ultimate supernatant was subjected to centrifugation at a temperature of 4 °C and a force of 12000 g for a duration of 30 minutes. The protein concentration of the resulting supernatant was determined using the BCA (Bicinchoninic acid) Protein Quantification Kit (Cat: 20201ES90, YEASEN, Shanghai, China), with fat-free BSA serving as a control. The protein samples that were obtained were subsequently subjected to an immunoblotting assay. Equal quantities of total protein extracted from the specified cells or liver samples were prepared for analysis using 10% or 12% SDS/PAGE gel (Cat: 20328ES50, SDS/PAGE Gel Preparation Kit, YEASEN, Shanghai, China). Subsequently, the proteins were transferred onto a 0.45 μM Immun-Blot polyvinylidene fluoride (PVDF) membrane (Cat: 10600023, Amersham Hybond, GE Healthcare Life Science, Germany) using the wetting transfer method. The transferred proteins were then probed with the designated primary antibodies in a western blotting procedure. Following this, the immunoblotting membranes were subjected to incubation with a blocking buffer consisting of 5% nonfat-dried milk (Cat: LP0033B, Biosharp® Life Science, Beijing, China) in a 1×TBS working buffer solution (Cat: PH1402, Scientific Phygene®) containing 0.1% Tween-20 (Cat: 9005-64-5, Sinopharm Chemical Reagent Co., Ltd., China) (referred to as 1×TBST working buffer solution) for a duration of 1 h. Subsequently, the membranes were combined with the designated primary antibodies and refrigerated at 4 °C overnight. Subsequently, the PVDF membranes underwent a triple wash in a 1×TBST working buffer solution. They were then subjected to co-treatment with horseradish peroxidase-tagged goat anti-rabbit IgG (H + L) or anti-mouse IgG (H + L) (Cat: 33201ES60; Cat: 33101ES60, YEASEN, Shanghai, China) for a duration of 1 to 2 h at a temperature range of 25 to 30 degrees Celsius. The visualisation of immunoblotting membranes was performed using the New-SUPER (Hypersensitivity Type) ECL Kit (Cat: KGP1128, KeyGen BioTECH, China) and the resulting images were captured using FUJI Medical X-ray film (Cat: 4741023952, FUJIFILM, China). Protein levels were subsequently determined by calculating the grey-scale score using Version 1.8.0 of the Image J software, developed by the National Institutes of Health (NIH) in the United States. These protein levels were then normalised to the expression of GAPDH and standardised as a fold change relative to the control samples.

## Statistical analysis

The results presented in this study were independently processed a minimum of three times. The raw data pertaining to this study were analysed independently using appropriate statistical techniques, as specified in the figure legends. Unless explicitly specified, numerical data scores are reported as the mean value accompanied by the standard error of the mean (SEM). The analysis of variance (ANOVA) was employed as a method of comparison, with several groups being subjected to *Dunnett's* multiple comparison test. Additionally, a two-group comparison was conducted using a two-tailed Student's *t* test. The final data presentation involved the use of various software tools. Specifically, GraphPad Prism Software (Version 9.4.1.681 for Microsoft Windows; GraphPad Software, San Diego, USA), Image (Version J 1.52g, NIH), R packages, and IBM SPSS Statistics (Version 25.0, Microsoft Windows, IBM, USA) were employed. A *P* value of less than 0.05 indicates the presence of a statistically significant difference.

## Reporting summary

Further information on research design is available in the Nature Portfolio Reporting Summary linked to this article.

## Data availability

The data that support the findings of this work are available from the Supplementary Information and Source data. There are no restrictions on data availability in the current work. Source data are provided with this paper.

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

## Acknowledgements

This work was supported by (1) National Natural Science Foundation of China (NSFC Grant No.: 82200652); (2) Chongqing Research Program of Basic Research and Frontier Technology (Grant No.: cstc2017jcyj-jAX0356, cstc2018jcyjA3686, cstc2018jcyjAX0784, cstc2018jcyjA1472, cstc2018jcyjAX0811, cstc2018jcyjA3533 & KJZDM201801601); (3) School-level Research Program of Chongqing University of Education (Grant No.: KY201710B & 17GZKP01); (4) Advanced Programs of Post-doctor of Chongqing (Grant No.: 2017LY39); (5) Science and Technology Research Program of Chongqing Education Commission of China (Grant No.: KJQN201901608, KJQN201901615, KJ1601402, KJZD-K202001603); (6) Children's Research Institute of National Center for Schooling Development Programme and Chongqing University of Education (Grant No.: CSDP19FSO1108); (7) Chongqing Professional Talents Plan for Innovation and Entrepreneurship Demonstration Team (Grant No.: CQCY201903258, cstc2021ycjh-bgzxm0202); (8) Natural Science Foundation of Chongqing, China (Grant No.: CSTB2022NSCQ-LZX0053, CSTB2022NSCQ-MSX0947 & CSTB2022NSCQ-MSX5364) and (9) Chongqing Entrepreneurship and Innovation Support Program for Overseas Returnees (Grant No.: CX2023011). We thank Dr. Yu-Ting Qin for critical reading of the manuscript. We also thank the members of the Prof. Yan-Rong Ren and Prof. Yuan-Yuan Li laboratory for their helpful discussions and insights on the manuscript.

## Author contributions

Conceptualisation & Methodology: M.X., J.T., Xin Liu, Li Han, B.W., J.C., Lianyi Han, S.S.; Investigation: M.X., J.T., X.L., Li Han, C.G., Y.Z., F.L, Z.W., X.X., L.X., X.W.g, Q.Z., X.W., Q.T., S.Z., Q.M., X.D., Q.K., Q.L., D.L., Linfeng Hu, Xi Liu, G.K., J.L., C.C., B.W., J.C., Lianyi Han, S.S.; Data analysis: M.X., J.T., Xin Liu, Li Han, S.Z., C.G., Lianyi Han; Funding acquisition, Project administration & Supervision: M.X., J.T., B.W., Lianyi Han, S.S.; Writing-original draft: M.X., J.T., C.G., B.W. Writing-review & editing: M.X., B.W., J.T., Lianyi Han, S.S.

## Competing interests

The authors declare no competing interests.
