## [Peer Review File · Nature Communications]

Reviewers' Comments:

Reviewer #1:

Remarks to the Author:

Thank you for allowing me the opportunity to review this important and timely work. Candidates for therapies in NASH, as the authors submit, are needed and potential targets that affect metabolic and fibrotic pathways are particularly interesting. The authors' rationale for choosing TRIM26 as a potential target is good, given biological plausibility as it is involved in metabolism and immunity. With regards to antifibrotic effects, data from ref 14 is limited, but does reference human data.

Main criticisms:

1. The impact of this work lies in the potential translation of preclinical observations into the clinical (human disease) phenotype. It is unclear from the description of methods used that human samples were systematically characterized, phenotyped and scored according to accepted criteria. The only pharmacologic therapy commented on were statins, pioglitazone and insulin, though many other medications (and subjects type 2 diabetes, lipid metabolism, etc) are potentially involved in the pathways affected by TRIM26.
2. It is unclear what the authors are trying to associate with the innate immune response effects and the known pathophysiology in NASH (references 11 and 15). This needs clarity.

Reviewer #2:

Remarks to the Author:

Dear Authors,

Thank you for submitting your manuscript. I very much enjoyed reading it.

You have found an inverse correlation between TRIM26 expression and NASH development. To prove a causal relationship you have constructed TRIM26 KO mice and conditional TRIM26 transgenic mice and showed that TRIM26 overexpression inhibits WTDF-induced steatosis and inflammation whereas TRIM26 deficiency, increased WTDF-induced steatosis and inflammation. You subsequently identified Cebpd as a target protein and showed with the creation of liver specific double KO's that all of the NASH phenotypes that are facilitated by Trim26 deficiency are alleviated by deficiency of Cebpd expression. You further showed in LO2 cells and primary hepatocytes that TRIM26 expression resulted in polyubiquitinated CEBPD.

Finally, as a proof of principle that TRIM26 could be used as gene therapy in NASH patients, you showed in rabbits that LV-Trim26 injection via liver portal vein downregulated dyslipidemia, hepatic steatosis, and hepatic injury in comparison to LV-Control groups after 8 weeks of HFHC diet treatment.

The manuscript contains an incredible amount of data and it took me a lot of time to through the supplements but it was worth the effort. Congratulations.

Minor comment: Please describe in the material and methods section how you performed the NAS score

Your sincerely,

Reviewer #3:

Remarks to the Author:

Overall, this manuscript is very interesting, the experimental design is well conducted, and the topic is quite relevant. The main goal of this study is to molecular mechanisms underlying the ability of Tripartite motif-containing protein 26 (TRIM26) in mitigating nonalcoholic steatohepatitis (NASH). The Authors pointed out that TRIM26 as potential therapeutic target for NASH as it exerts a critical role in the suppression of CCAAT/enhancer binding protein delta (C/EBPdelta). Indeed,

trim26 catalyzes the ubiquitination of C/EBPdelta in hepatocytes, thus priming its consequent degradation and suppressing NF- κ B p65 activation. Hepatocyte-specific TRIM26 genetic ablation favors inflammation and fibrosis, exacerbating severe NASH. In keeping with these findings, trim26 restoration ameliorates NASH in preclinical models. The results are deeply validated in several models and deemed of interest.

The Authors might be willing to address the following points:

- The rationale of the study needs to be clarified. Still, it appears that the motivation of this study has been superficially justified in both abstract and main text.
- The introduction section is quite accurate, appropriately citing important contributions in this field. The main purposes of the manuscript are clearly stated only in the methods sections.
- In the Methods section please clarify the sentence 'The relevant non-steatotic liver tissues were obtained from donors who were not eligible for liver transplantation for non-liver'
- Study design: is well explained and rigorously conducted. The authors replicate their findings in human livers, derived from patients at different stages of the disease, in a wide series of preclinical models and in in vitro.
- The methods section is largely descriptive, and appropriate. The statistical analyses are well described and multiple comparisons analyses have been addressed.
- I suggest to provide the clinical features of patients included in the study directly into the main text, presenting mean, standard deviation and statistical analysis, so that they are more immediately suitable for the readers. How the Authors define NASH?
- Did the Authors try to perform multivariate analysis for patients' data? The Authors may try to correct their analysis adjusting for clinical features and medications.
- In the Results section, the Authors might have to report all statistical analysis, comparison and p values of the analyses. The same observations should be applied also to figure legends.
- The Discuss section is accurate and includes the comparisons with important contributions in this field.

Minor revisions:

- Rephrase the lines 112-113
- Rephrase the lines 159-160
- Line 195 NASH pathology \diamond NASH
- Line 819-820 \diamond invalid characters
- Rephrase the lines 498-500
- Please check all abbreviation in the abstract and in the main text.
- Authors should improve the drafting of all paper and the use of English language should be checked.
- Authors should check typos in text.

Reviewer #4:

In this study, authors demonstrated that TRIM26 is regulated by HNF4a and interacts with CEBPD to catalyze the ubiquitination and degradation of CEBPD in hepatocytes. Hepatocyte-specific *Trim26* deletion significantly promoted the progression of NASH-related phenotypes. In contrast, *Trim26* overexpression in transgenic mice, lentivirus (LV) or adeno-associated virus (AAV) -induced Trim26 gene therapy attenuated NASH-associated phenotypes. TRIM26 directly interacted with CEBPD and promoted the degradation of ubiquitin-proteasome, thereby inhibiting the activation of CEBPD-HIF1a-NF- κ B-p65 signaling pathway and its downstream pathways. This study used a large number of animal models and transgenic mice, which is full of content, however there are still some of the following questions need to be answered.

Major concerns:

1. Fig. 1 mainly illustrated the correlation between TRIM26 and severity of NASH in mouse NASH models and clinical NASH samples:
 - 1) Among the four candidate genes, only *Trim8* showed a significant increase while the other three showed decrease in mRNA levels, and it seemed more meaningful to select *Trim8*, why chose *Trim26* as the subject?
 - 2) As seen in the peak graphs of Fig. 1C, TRIM26 level in L02 cells was low at the initial stage, transiently increased and then decreased again. The initial low level does not seem to be consistent with the conclusions of the article since low expression of *Trim26* promoted the NASH. Authors should provide experimental validation results (qPCR for *Trim26*) for sequencing data and explain this phenomenon.
 - 3) For TRIM26 protein level detected in Fig. S2, authors should select the same time point as in Figure C for detection.
 - 4) Figure D showed the expression of TRIM26 and CEBPD in clinical samples of NASH but lack of evaluation for NASH, relevant data (HE staining, immunohistochemical staining, F4/80, *etc.*) of clinical section should be performed and statistical analysis of the correlation between TRIM26, CEBPD and NASH should be done as well.

5) The correlation between TRIM26 and CEBPD shown in Figure E, as well as the results of RNA-seq for L02 in Figure 2 are all at mRNA level, while in this study TRIM26's effect on CEBPD was based on the regulation of ubiquitination. Therefore, these data can only indicate the negative correlation of *Trim26* and *cebpd* at RNA level, having no inspiration on ubiquitination for subsequent research. Authors should perform a proteome spectrum analysis to find ubiquitinating substrates of TRIM26.

2. Alb-Cre mainly knocked out TRIM26 in hepatocytes while having no effect on TRIM26 in nonparenchymal hepatic cells, theoretically some TRIM26 could be detected in WB. However, the presence of TRIM26 in the knockout group was blank in Fig. S4B-4C, authors should perform immunohistochemistry staining or immunofluorescence staining for further detection of knockout effect.

3. Co-IP experiments in Fig. 6F used CEBPD to pull down TRIM26 twice, in addition, the bands of TRIM26 (63kD) and CEBPD (25kD) have same molecular weight in the Input of GST-pull down assay. Data shown in Fig. 6 cannot fully confirm the interaction between CEBPD and TRIM26. Authors should perform co-IP by using TRIM26 to pull down CEBPD and GST-pull down assay to prove the interaction.

4. Why do Fig 6G, CEBPD-▲-BLZ-Flag Input showed three bands and CEBPD-▲-LZR-Flag Input showed two bands?

5. In Fig7G, lane 4 was not transfected with Myc-Ub, whereas the degradation zone appeared in IB: Myc. Authors should design and perform this experiment again.

6. What is the advantages of rabbits over the use of mice as experimental subjects in Fig. 8?

Minor concerns:

1. Multiple animal models were used in this study, and tissue staining showed that NASH was aggravated after TRIM26 knockdown. Authors should show the representative livers of disease to prove whether liver changed in appearance.
2. WB results in this research can show the expression trend correctly, but almost all WB results have the phenomenon of tailing, which may be caused by high temperature when transferring or excessive residual fat in the process of protein extraction. It is need to extract the protein for WB again, so as to facilitate the display of the results more clearly and effectively.
3. All proteins should be capitalized, such as TRIM26, CEBPD (Fig.6A,6B, Fig.7C.7E, etc.)
4. Appropriate magnification should be applied in the immunofluorescent staining according to Fig. 6I
5. In Fig. S3, 'AdTrim26-treasfected' was wrong written, it should be AdTrim26-transfected.
6. English should be improved by native speaking professional editor.

RESPONSE TO THE REVIEWER COMMENTS (NCOMMS-23-00745)

According to comments and suggestions, we improved this study. All changes made in the manuscript are marked as **red**.

Reviewer #1 (Remarks to the Author):

Thank you for allowing me the opportunity to review this important and timely work. Candidates for therapies in NASH, as the authors submit, are needed and potential targets that affect metabolic and fibrotic pathways are particularly interesting. The authors' rationale for choosing TRIM26 as a potential target is good, given biological plausibility as it is involved in metabolism and immunity. With regards to antifibrotic effects, data from ref 14 is limited, but does reference human data.

We appreciate the reviewer's compliment on our manuscript's importance that "*The authors' rationale for choosing TRIM26 as a potential target is good, given biological plausibility as it is involved in metabolism and immunity.*" We have worked hard to address all of the comments made by this reviewer.

Main criticisms:

Question 1:

1. The impact of this work lies in the potential translation of preclinical observations into the clinical (human disease) phenotype. It is unclear from the description of methods used that human samples were systematically characterized, phenotyped and scored according to accepted criteria. The only pharmacologic therapy commented on were statins, pioglitazone and insulin, though many other medications (and subjects type 2 diabetes, lipid metabolism, etc) are potentially involved in the pathways affected by TRIM26.

Response 1:

We thank the reviewer for this comment, as indicated in the manuscripts, the human liver samples used in the current work were obtained from our previous study (**Ref 1, indicated below**). Notably, the human donors' liver specimens were harvested from adult donors with NAFLD who underwent biopsy tissue samples or liver transplantation samples. The relevant non-steatotic liver tissues were obtained from donors who were not eligible for liver transplantation for non-liver reasons. Non-steatosis samples, Simple steatosis samples, and NASH phenotype liver samples were obtained and included in this study. Of note, steatotic liver samples from patients with any of the following conditions were excluded from the study: excessive drinking (alcohol >70 g for female or alcohol >140 g for male, per week), viral infection or drug abuse (including hepatitis B & C virus infection). Samples from non-steatotic liver were collected from the normal region of the livers from donors who received liver resection owing to liver hemangioma or hepatic cyst. Hierarchical steatosis and steatohepatitis were independently diagnosed by two pathologists according to the scoring system of standard histological criteria established by the NASH Clinical Research Network (Ref 2, indicated below). Cases with NAFLD activity scores (NAS) of 1–2, and ballooning scores of 0 and no fibrosis, were classified as simple steatosis. Cases with NAS \geq 5 or NAS of 3–4 but with fibrosis were classified as NASH. Cases with NAS of 0 were classified as normal. Also, prior to this study, the samples of non-steatosis and simple steatosis in this cohort were collected from patients without taking statins or insulin. The NASH phenotype liver samples in this cohort were from patients who had taken pioglitazone (15-30 mg/day) for no more than 24 months. Liver sample donors and their families agree & sign written informed consent. All protocols involving human donors in this work were grounded on *the Ethical Principles for Medical Research Involving Human Subjects, Declaration of Helsinki (64th WMA general*

assembly), and totally approved by the *Academic Committee of Experimental Animal Ethics, Use & Care Union* in Chongqing University of Education and other participating units.

Refs:

1. Xu, Min-Xuan, et al. "Tripartite motif-containing protein 31 confers protection against nonalcoholic steatohepatitis by deactivating mitogen-activated protein kinase kinase kinase 7." *Hepatology* 77.1 (2023): 124-143.
2. Kleiner, D.E. et al. Design and validation of a histological scoring system for nonalcoholic fatty liver disease. *Hepatology* 41, 1313–1321 (2005).

Undoubtedly, in addition to statins, pioglitazone, and insulin, there are many other drugs associated with NAFLD complications such as obesity, diabetes, hyperlipidemia, and hypertension that may be affected by TRIM26. However, in view of the numerous complications of NAFLD, complicated pathology, and diverse clinical heterogeneity, we tried our best to exclude the factors that may be most affected by TRIM26 and the most widely used clinical drugs when selecting liver samples.

Question 2:

2. *It is unclear what the authors are trying to associate with the innate immune response effects and the known pathophysiology in NASH (references 11 and 15). This needs clarity.*

Response 2:

We fully agree with the reviewer's concern regarding the correlation of innate immune response effects with the known pathophysiology in NASH. Previous reports have confirmed that chronic liver inflammation induced by over-nutrition diet consumption is a common trigger of liver disease, and is considered the main driver of hepatic tissue damage leading to NASH and hepatocellular carcinoma (HCC). The inflammatory phenotype during liver injury can be attributed to the innate immune system, which is the first line of defense against invading pathogens and is crucial for the overall survival of the host. Liver innate immune cells include Kupffer cells, monocytes, neutrophils, dendritic cells (DCs), natural killer (NK) cells, and NK T (NKT) cells, and initiate and maintain hepatic inflammation through cytokine production (**Ref 1, 2 indicated below**). A dysregulated cytokine balance after liver injury can result in excessive cell death of hepatocytes, a key finding in various acute and chronic liver diseases (**Ref 3 indicated below**). Cytokines can activate effector functions of immune cells as well as hepatocytic intracellular signaling pathways, and thus play an important role in the interplay between intrahepatic immune cells and hepatocytes.

More importantly, due to the liver is a site where foreign antigens from the gastrointestinal tract encounter the immune system, it is particularly enriched with innate immune cells. These cells can modify and disrupt critical processes implicated in metabolic disease. As such, metabolic stress initiates a feedforward cycle of inflammatory responses, resulting in a state of unresolved chronic inflammation in the liver. Accordingly, the crosstalk between these innate immune cells and the resident parenchymal cells plays an important role in the development of acute and chronic liver disease (**Ref 3 indicated below**).

Of note, as we mentioned in Introduction section, TRIM26 has been determined as a multifunctional regulator in innate immune response and chronic metabolic diseases development by regulating the targeted substrate ubiquitination modification. These examples (*e.g.*, references 11 and 15) indicated that TRIM26 was widely involved in diseases pathological and physiological processes and possibly played unique functions in liver diseases. Therefore, the important role of TRIM26 in regulating different immune responses and the occurrence of diseases was further

demonstrated. Additionally, the function of TRIM26, especially in NASH pathogenesis, remains elusive, so these facts compel us to investigate the potential molecular mechanism of TRIM26.

Refs:

1. Liaskou, Evaggelia, Daisy V. Wilson, and Ye H. Oo. "Innate immune cells in liver inflammation." *Mediators of inflammation* 2012 (2012).
2. Schattenberg, Jorn Markus, Marcus Schuchmann, and Peter Robert Galle. "Cell death and hepatocarcinogenesis: Dysregulation of apoptosis signaling pathways." *Journal of gastroenterology and hepatology* 26 (2011): 213-219.
3. Bieghs, Veerle, and Christian Trautwein. "The innate immune response during liver inflammation and metabolic disease." *Trends in immunology* 34.9 (2013): 446-452.

Reviewer #2 (Remarks to the Author):

Dear Authors,

Thank you for submitting your manuscript. I very much enjoyed reading it.

You have found an inverse correlation between TRIM26 expression and NASH development. To prove a causal relationship you have constructed TRIM26 KO mice and conditional TRIM26 transgenic mice and showed that TRIM26 overexpression inhibits WTDF-induced steatosis and inflammation whereas TRIM26 deficiency, increased WTDF-induced steatosis and inflammation.

You subsequently identified Cebpd as a target protein and showed with the creation of liver specific double KO's that all of the NASH phenotypes that are facilitated by Trim26 deficiency are alleviated by deficiency of Cebpd expression. You further showed in LO2 cells and primary hepatocytes that TRIM26 expression resulted in polyubiquitinated CEBPD.

Finally, as a proof of principle that TRIM26 could be used as gentherapy in NASH patients, you showed in rabbits that LV-Trim26 injection via liver portal vein downregulated dyslipidemia, hepatic steatosis, and hepatic injury in comparison to LV-Control groups after 8 weeks of HFHC diet treatment.

The manuscript contains an incredible amount of data and it took me a lot of time to through the supplements but it was worth the effort. Congratulations.

Question :

Minor comment: Please describe in the material and methods section how you performed the NAS score.

Response:

We thank this reviewer for the strong support! NAFLD Activity Score evaluation used in our current study was accordance with our previous reports and published literature (**Ref 1, 2 indicated below**). The NAS is the sum of separate scores for steatosis (0-3), hepatocellular ballooning (0-2), and lobular inflammation (0-3).

The following table indicated the evaluation score.

Item	Definition	Score
Steatosis	<5%	0
	5-33%	1
	>33% to 66%	2
	>66%	3
Lobular inflammation	No foci	0
	<2 foci/×200 field	1
	2-4 foci/×200 field	2
	>4 foci/×200 field	3
Hepatocyte ballooning	None	0
	Few balloon cells	1
	Many cells/prominent ballooning	2

Refs:

1. Azushima, Kengo, et al. "Adipocyte-specific enhancement of angiotensin II type 1 receptor-associated protein ameliorates diet-induced visceral obesity and insulin resistance." Journal of the American Heart Association 6.3 (2017): e004488.
2. Xu, Min-Xuan, et al. "Tripartite motif-containing protein 31 confers protection against nonalcoholic steatohepatitis by deactivating mitogen-activated protein kinase kinase kinase 7." Hepatology 77.1 (2023): 124-143.

Reviewer #3 (Remarks to the Author):

Overall, this manuscript is very interesting, the experimental design is well conducted, and the

topic is quite relevant. The main goal of this study is to molecular mechanisms underlying the ability of Tripartite motif-containing protein 26 (TRIM26) in mitigating nonalcoholic steatohepatitis (NASH). The Authors pointed out that TRIM26 as potential therapeutic target for NASH as it exerts a critical role in the suppression of CCAAT/enhancer binding protein delta (C/EBPdelta). Indeed, trim26 catalyzes the ubiquitination of C/EBPdelta in hepatocytes, thus priming its consequent degradation and suppressing NF- κ B p65 activation. Hepatocyte-specific TRIM26 genetic ablation favors inflammation and fibrosis, exacerbating severe NASH. In keeping with these findings, trim26 restoration ameliorates NASH in preclinical models. The results are deeply validated in several models and deemed of interest.

We thank a lot for the reviewer's positive comments, insightful criticism and constructive suggestions which heavily strengthen our study. As such, we performed requested experiments by the reviewer and revised the manuscript to improve the clarity.

The Authors might be willing to address the following points:

Question 1:

The rationale of the study needs to be clarified. Still, it appears that the motivation of this study has been superficially justified in both abstract and main text.

The introduction section is quite accurate, appropriately citing important contributions in this field. The main purposes of the manuscript are clearly stated only in the methods sections.

Response 1:

Thanks for the reviewer's mention. According to your suggestion, we have reorganized our abstract part and some sections of main text to highlight our study purposes to follow your concerns. The purposes statement of our study have also been shifted to the last paragraph of Introduction to make our statement clear and complete.

Question 2:

In the Methods section please clarify the sentence "The relevant non-steatotic liver tissues were obtained from donors who were not eligible for liver transplantation for non-liver".

Response 2:

We thank the reviewer for this suggestion, as indicated in the manuscripts, non-steatotic liver was used as the control. As clarified in the Method section that "Samples from non-steatotic liver were collected from the normal region of the livers from donors who received liver resection owing to liver hemangioma or cyst of liver." Of note, all donor livers were allocated via China Organ Transplant Response System from 2016 to 2022. The donors were enrolled in the study on a volunteer basis, and the families of organ donors were approached for consent. Written informed consent was obtained from subjects or families of all participants.

Question 3:

Study design: is well explained and rigorously conducted. The authors replicate their findings in human livers, derived from patients at different stages of the disease, in a wide series of preclinical models and in in vitro.

The methods section is largely descriptive, and appropriate. The statistical analyses are well described and multiple comparisons analyses have been addressed.

I suggest to provide the clinical features of patients included in the study directly into the main text, presenting mean, standard deviation and statistical analysis, so that they are more immediately suitable for the readers. How the Authors define NASH?

Response 3:

We thank this reviewer for the strong support! We have provided the clinical features of patients included in our present study directly into the manuscript to follow your comments (Table 1, indicated below).

Table 1. The clinical information of non-steatosis, simple steatosis and NASH patients.

Variable	Non-steatosis	Simple steatosis	NASH
Sex	Male/Female (11/5)	Male/Female (10/7)	Male/Female (8/8)
Age (y)	35.06±8.00	35.70±8.54	35.93±10.94
BMI (Kg/m ²)	19.55±2.47 ^{a, b}	25.10±1.32 ^b	30.04±2.84
Serum AST (U/L)	24.79±7.53 ^{a, b}	54.18±8.48	51.93±9.52
Serum ALT (U/L)	22.21±8.46 ^{a, b}	45.50±7.62	42.06±8.67
Serum TG (mmol/L)	1.71±0.28 ^{a, b}	2.13±0.24 ^b	2.27±0.42
Serum TC (mmol/L)	3.76±0.48 ^{a, b}	4.17±0.43 ^b	4.42±0.85
Serum FBG (mmol/L)	5.03±0.75 ^{a, b}	5.71±0.70 ^b	6.27±0.77
Liver IL6 mRNA	1.55±0.24 ^{a, b}	2.45±0.27 ^b	3.48±0.57
Liver TRIM26 mRNA	0.95±0.30 ^{a, b}	0.52±0.17 ^b	0.27±0.13
Liver TNF mRNA	1.39±0.31 ^{a, b}	2.37±0.39 ^b	4.45±0.39
Liver CEBPD mRNA	0.95±0.26 ^{a, b}	3.99±1.93 ^b	5.76±1.41
Serum LN (µg/L)	78.27±28.65 ^{a, b}	92.56±25.89 ^b	153.84±40.36
Serum HA (µg/L)	36.97±11.24 ^b	31.95±13.51 ^b	105.06±26.23
Serum IVC (µg/L)	70.93±25.75 ^b	71.01±30.76 ^b	111.39±26.72
Serum PCIII (µg/L)	49.25±22.90 ^{a, b}	52.84±24.95 ^b	120.00±41.23
Serum GGT (U/L)	28.91±10.82 ^{a, b}	36.68±18.61 ^b	68.39±16.52
Serum AKP (U/L)	86.69±19.32 ^{a, b}	106.84±30.55 ^b	181.33±37.51
NAS score	0.00±0.00 ^{a, b}	1.64±0.49 ^b	4.31±0.70

These data are expressed as the mean ± SEM. ^aP<0.05 versus the Simple steatosis donors; ^bP<0.05 versus the NASH donors. Abbreviation: BMI, Body mass index; AST, Aspartate transaminase; ALT, Alanine aminotransferase; TG, Triglyceride; TC, Total cholesterol; FBG, Fasting blood glucose; IL6, interleukin-6; TRIM26, Tripartite motif containing 26; TNF, Tumor necrosis factor; CEBPD, CCAAT/enhancer binding protein delta; LN, Laminin; HA, hyaluronidase; IVC, Collagen Type IV; PCIII, Type III procollagen; GGT, γ-glutamyl transpeptidase; AKP, Alkline phosphatase

We thank the reviewer for this comment, as indicated in the manuscripts, the human liver samples used in the current work were obtained from our previous study (**Ref 1, indicated below**). Notably, the human donors' liver specimens were harvested from adult donors with NAFLD who underwent biopsy tissue samples or liver transplantation samples. The relevant non-steatotic liver tissues were obtained from donors who were not eligible for liver transplantation for non-liver reasons. Samples from non-steatotic liver were collected from the normal region of the livers from donors who received liver resection owing to liver hemangioma or hepatic cyst. Hierarchical steatosis and steatohepatitis were independently diagnosed by two pathologists according to the scoring system of standard histological criteria established by the NASH Clinical Research Network (Ref 2, indicated below). Cases with NAFLD activity scores (NAS) of 1–2, and ballooning scores of 0 and no fibrosis, were classified as simple steatosis. Cases with NAS ≥ 5 or NAS of 3–4 but with fibrosis were classified as NASH. Cases with NAS of 0 were classified as normal.

Refs:

1. Xu, Min-Xuan, et al. "Tripartite motif-containing protein 31 confers protection against nonalcoholic steatohepatitis by deactivating mitogen-activated protein kinase kinase 7." *Hepatology* 77.1 (2023): 124-143.
2. Kleiner, D.E. et al. Design and validation of a histological scoring system for nonalcoholic fatty liver disease. *Hepatology* 41, 1313–1321 (2005).

Question 4:

Did the Authors try to perform multivariate analysis for patients' data? The Authors may try to correct their analysis adjusting for clinical features and medications.

Response 4:

Thank you so much for your concerns here. According to your comments, we have performed the multiple regression analysis for patients' data (indicated in Supplementary table 1).

The PCA analysis and multiple regression analysis (correlation) are indicated below:

PCA analysis

Multiple regression analysis (correlation)

Also, we additionally performed the H&E staining for our clinical samples to detect histological change. Meanwhile, the corresponding NAS score for our clinical samples are also attached to facilitate better comparison.

This result has been updated in Revised Supplementary figure 1.

These data consistently demonstrate that serum indexes associated with NASH progression in patients are negatively correlated with TRIM26 expression.

Question 5:

In the Results section, the Authors might have to report all statistical analysis, comparison and p values of the analyses. The same observations should be applied also to figure legends. The Discuss section is accurate and includes the comparisons with important contributions in this field.

Response 5:

Thank you very much for your suggestions here. We have indicated the statistical analysis, and corresponding *P* values in results section and legends. We also appreciate the reviewer's compliment on our manuscript's importance regarding statement in discussion section.

Question 6:

*Minor revisions:
-Rephrase the lines 112-113*

- Rephrase the lines 159-160*
- Line 195 NASH pathology □ NASH*
- Line 819-820 □ invalid characters*
- Rephrase the lines 498-500*
- Please check all abbreviation in the abstract and in the main text.*
- Authors should improve the drafting of all paper and the use of English language should be checked.*
- Authors should check typos in text.*

Response 6:

Thank you very much for the reviewer's comments here. We have revised the whole manuscript to correct the corresponding issues you mentioned above. The revised sections or sentences have been marked in our revised file for your consideration. We hope these changes meet your requirements.

Reviewer #4 (Remarks to the Author):

In this study, authors demonstrated that TRIM26 is regulated by HNF4a and interacts with CEBPD to catalyze the ubiquitination and degradation of CEBPD in hepatocytes. Hepatocyte-specific Trim26 deletion significantly promoted the progression of NASH-related phenotypes. In contrast, Trim26 overexpression in transgenic mice, lentivirus (LV) or adeno-associated virus (AAV) -induced Trim26 gene therapy attenuated NASH-associated phenotypes. TRIM26 directly interacted with CEBPD and promoted the degradation of ubiquitin-proteasome, thereby inhibiting the activation of CEBPD-HIF1 α -NF- κ B-p65 signaling pathway and its downstream pathways. This study used a large number of animal models and transgenic mice, which is full of content, however there are still some of the following questions need to be answered.

We thank this reviewer for the strong support! We have worked hard to address all of the comments made by this reviewer.

Major concerns:

Question 1-1:

1. Fig. 1 mainly illustrated the correlation between TRIM26 and severity of NASH in mouse NASH models and clinical NASH samples:

1) Among the four candidate genes, only Trim8 showed a significant increase while the other three showed decrease in mRNA levels, and it seemed more meaningful to select Trim8, why chose Trim26 as the subject?

Response 1-1:

Thank you so much for your concerns here. The joint methods were used here to screen for E3 ubiquitin ligases that vary widely in expression. Actually, TRIM8 has been widely confirmed to be the potential regulatory E3 ubiquitin enzyme in NASH development, and its mechanism has been studied previously (**Ref 1, 2, indicated below**), which can be used as a promising therapeutic target. It further proves that our screening strategy in present study is feasible and effective.

Refs:

1. Yan, Feng-Juan, et al. "The E3 ligase tripartite motif 8 targets TAK1 to promote insulin resistance and steatohepatitis." *Hepatology* 65.5 (2017): 1492-1511.
2. Chen, Ze, et al. "Emerging molecular targets for treatment of nonalcoholic fatty liver disease." *Trends in Endocrinology & Metabolism* 30.12 (2019): 903-914.

Of note, TRIM26 expression is severely downregulated in the setting of NASH, and its role in NASH progression has not been reported so far, so we attempt to study it and uncover its biological function.

Question 1-2:

2) As seen in the peak graphs of Fig. 1C, TRIM26 level in L02 cells was low at the initial stage, transiently increased and then decreased again. The initial low level does not seem to be consistent with the conclusions of the article since low expression of Trim26 promoted the NASH. Authors should provide experimental validation results (qPCR for Trim26) for sequencing data and explain this phenomenon.

Response 1-2:

We fully agree with the reviewer on this issue, because this dramatic reduction of *TRIM26* mRNA level occurs only at a very early stage (0-1 h) and seems to occur only in human L02 cells, no similar observation has been found in mouse primary hepatocytes. Therefore, we suspect that this may be due to the acute cellular stress response of L02 cells when suddenly stimulated by PAOA, which also indicates that L02 cells may be more sensitive to lipotoxic stimuli. As such,

considering the confirmation of hepatocyte nuclear factor 4-alpha (HNF4A) as a key promoter-binding transcription factor for TRIM26 expression, we next analyzed the TRIM26 transcriptional activity by HNF4A using luciferase reporter in the very early stage (0-1 h) and over the next few hours (1-8 h) in both L02 cells.

Using methods similar to those in Supplementary figure 1, as expected in the following results, at a very early stage (0-1 h), the acute cellular stress response induced by PAOA significantly reduced the binding of the transcription factor HNF4A, which also directly led to reduced transcriptional expression of TRIM26 (left). Subsequently, intracellular HNF4A resumed the transcription of TRIM26 in response to the continuous stimulation of PAOA, resulting in a temporary upregulation in TRIM26 mRNA expression. Since the lipid toxicity-mediated by PAOA can significantly inhibit the expression of HNF4A (*Ref 1, 2, indicated below*), the subsequent reduction of HNF4A expression further reduces the expression of TRIM26. At the same time, depletion of the existing TRIM26 protein also aggravated the decrease in TRIM26 expression. This phenomenon was also confirmed at the dynamic alteration in TRIM26 mRNA level (right).

Refs:

1. Liu, Dan, et al. "TNFAIP3 interacting protein 3 overexpression suppresses nonalcoholic steatohepatitis by blocking TAK1 activation." *Cell metabolism* 31.4 (2020): 726-740.
2. Ye, Ping, et al. "Dual-specificity phosphatase 26 protects against nonalcoholic fatty liver disease in mice through transforming growth factor beta-activated kinase 1 suppression." *Hepatology* 69.5 (2019): 1946-1964.

Meanwhile, we also detected the *TRIM26* mRNA expression in 0-32 h during PAOA treatment in mouse primary hepatocytes and HepG2 cells.

Compared with the results found in L02 cells, only slight TRIM26 mRNA expression fluctuations were observed in HepG2 and mouse primary hepatocytes at the early stage of PAOA treatment, which further indicated that this phenomenon may only exist in a few cell types, such as L02 cells, or that L02 cells may be more sensitive to PAOA treatment.

Question 1-3:

3) For TRIM26 protein level detected in Fig. S2, authors should select the same time point as in Figure C for detection.

Response 1-3:

Thank you very much for your comments. Here we re-performed the western blotting analysis for TRIM26 protein expression in 0 h, 8 h, 16 h, and 32 h during metabolic insult challenge. This result has been updated in revised Supplementary figure 2.

Representative immunoblotting bands of TRIM26 in the 0.5 mM palmitic acid+1.0 mM oleic acid (PAOA) mixture (A, B), 5 mM fructose or 100 ng/ml TNF- α (C)-induced WT L02 cells for time-course treatment or 32 hours treatment ($n=4$ per group).

Question 1-4:

4) Figure D showed the expression of TRIM26 and CEBPD in clinical samples of NASH but lack of evaluation for NASH, relevant data (HE staining, immunohistochemical staining, F4/80, etc.) of clinical section should be performed and statistical analysis of the correlation between TRIM26, CEBPD and NASH should be done as well.

Response 1-4:

We fully agree with the reviewer on this point, according to your comments, we additionally performed the H&E staining for our clinical samples to detect histological change. Meanwhile, the corresponding NAS score for our clinical samples are also attached to facilitate better comparison. This result has been updated in Revised Supplementary figure 1.

These data consistently demonstrate that clinical indexes associated with NASH progression in patients are negatively correlated with TRIM26 expression.

Question 1-5:

5) The correlation between TRIM26 and CEBPD shown in Figure E, as well as the results of RNA-seq for L02 in Figure 2 are all at mRNA level, while in this study TRIM26's effect on CEBPD was based on the regulation of ubiquitination. Therefore, these data can only indicate the negative correlation of Trim26 and cebpd at RNA level, having no inspiration on ubiquitination for subsequent research. Authors should perform a proteome spectrum analysis to find ubiquitinating substrates of TRIM26.

Response 1-5:

Thank you so much for your constructive points here. We fully agree with the reviewer on this issue, thus according to your comments, with the aim of identifying the substrate that was directly regulated by TRIM26 during fatty liver disease, we coimmunoprecipitated TRIM26 with its interacting proteins and then performed mass spectrometry in human L02 hepatocytes that were incubated with PAOA. The method used in this experiment was in accordance with our previous report and published literature with certain modifications (*Refs 1&2*, indicated below).

Briefly, the L02 hepatocytes were transfected with HA-tagged *TRIM26* using Thermo Scientific TurboFect reagent (#R0531). After incubation with PAOA or BSA control for 10 h, cell samples were lysed and collected from the 2 experimental groups. The corresponding TRIM26 complexes were subsequently immunoprecipitated as described in the IP assay. The harvested TRIM26 complexes were digested in solution with trypsin solution (#V5280; Promega). The peptides were extracted twice using 1% trifluoroacetic acid in a 50% acetonitrile aqueous solution for 30 min. The extracts were then centrifuged in a SpeedVac vacuum concentrators to reduce the liquid volume. To perform the liquid chromatography–tandem mass spectrometry (LC–MS/MS) detection, the digestion samples were separated using a 2.5 h gradient elution at a flow rate of 0.300 μ l/min in an UltiMate 3000™-HPLC (Thermo Scientific™) platform that was directly interfaced with a Thermo Q Exactive mass spectrometer (Thermo Scientific™). The analytical column consisted of a fused-silica capillary column (75 μ m inner diameter, 150 mm length; Upchurch, Oak Harbor, WA) packed with C-18 resin (300 \AA , 5 μ m; Varian, Lexington, MA). Mobile phase A consisted of 0.1% formic acid in water, and mobile phase B consisted of 0.1% formic acid in 100% acetonitrile. A Q Exactive mass spectrometer was operated in data-dependent acquisition mode using Xcalibur 2.2 software with a single full-scan mass spectrum in an Orbitrap followed by ten data-dependent MS/MS scans in an ion trap at 35% of the normalized collision energy. For protein identification, Proteome Discoverer 2.1 software (Thermo Fisher Scientific) was used for database searches for peptide mass fingerprinting and peptide sequence tagging.

As expected in our mass spectrometry analysis, consistent with our other results, the CEBPD is a major substrate of TRIM26 during metabolic insult challenge.

Refs:

3. Xu, Min-Xuan, et al. "Tripartite motif-containing protein 31 confers protection against nonalcoholic steatohepatitis by deactivating mitogen-activated protein kinase kinase 7." *Hepatology* 77.1 (2023): 124-143.
4. Wang, Lin, et al. "Tripartite motif 16 ameliorates nonalcoholic steatohepatitis by promoting the degradation of phospho-TAK1." *Cell Metabolism* 33.7 (2021): 1372-1388.

Question 2:

2. *Alb-Cre* mainly knocked out *TRIM26* in hepatocytes while having no effect on *TRIM26* in nonparenchymal hepatic cells, theoretically some *TRIM26* could be detected in WB. However, the presence of *TRIM26* in the knockout group was blank in Fig. S4B-4C, authors should perform immunohistochemistry staining or immunofluorescence staining for further detection of knockout effect.

Response 2:

Thank you very much for your comments here. According to your concerns regarding our results in Supplementary figure 4B and 4C, we first rechecked our raw western blotting data, and speculated that it might be due to X-ray film scanning and exposure that caused the phenomenon you mentioned above.

Supplementary figure 4B&4C

Trim26 raw WB bands in Supplementary figure 4B&4C

Furthermore, to determine the *Trim26*-specific knockout in hepatocytes, as indicated below, we directly detected the *Trim26* protein expression in primary hepatocytes and the other cell types components isolated from *Trim26*-HKO mice.

As expected, indeed, this knockout strategy used in our work only deleted Trim26 expression in hepatocytes, and other cell types were not affected.

Question 3:

3. Co-IP experiments in Fig. 6F used CEBPD to pull down TRIM26 twice, in addition, the bands of TRIM26 (63kD) and CEBPD (25kD) have same molecular weight in the Input of GST-pull down assay. Data shown in Fig. 6 cannot fully confirm the interaction between CEBPD and TRIM26. Authors should perform co-IP by using TRIM26 to pull down CEBPD and GST-pull down assay to prove the interaction.

Response 3:

Thank you so much for your comments here. We have re-performed this experiment to further determine the direct interaction between TRIM26 and CEBPD. The Thermo Scientific™, Pierce® Classic Magnetic IP/Co-IP Kit (Cat#: 88804) and Pierce® GST Protein Interaction Pull-Down Kit (Cat#: 21516) were used to perform this experiment.

As expected, the co-IP and GST pull down experiment again confirmed that CEBPD and TRIM26 directly interact with each other. The corresponding result has been updated in revised figure 6.

Question 4:

4. Why do Fig 6G, CEBPD- Δ -BLZ-Flag Input showed three bands and CEBPD- Δ -LZR-Flag Input showed two bands?

Response 4:

Thank you for your comments here. The reason for this phenomenon may be attributed to the non-specific binding of the protein. We repeated this experiment using the same kit, which was consistent with our other results and further identified the major binding domain of CEBPD and TRIM26. This result has been updated in revised figure 6.

Question 5:

5. In Fig7G, lane 4 was not transfected with Myc-Ub, whereas the degradation zone appeared in IB: Myc. Authors should design and perform this experiment again.

Response 5:

Thank you so much for your issue in this regard, we have re-performed this experiments using the same protocol as we described previously. Consistently, we reconfirmed the effect of TRIM26 on CEBPD via increase in ubiquitination levels of CEBPD. This result has been updated in revised figure 7.

Question 6:

6. What is the advantages of rabbits over the use of mice as experimental subjects in Fig. 8?

Response 6:

It is very difficult to investigate the molecular pathogenesis of NAFLD and NASH in humans because of the heterogeneity of human populations and wide differences in their diet and lifestyle. More importantly, the inability to obtain multiple liver biopsies from patients with NAFLD or NASH and healthy volunteers adds to the difficulty of these studies. Given these technical issues and associated ethical challenges involved in studying patients with NAFLD, considerable effort has been expended to develop animal models of fatty liver disease.

Ideally, an animal model to study NASH should replicate three important phenotypic characteristics of human disease. First, the animal should have the characteristic metabolic abnormalities, namely, insulin resistance, hyperglycemia, hyperlipidemia, and visceral adiposity. Second, the animals should develop lipid accumulation within hepatocytes along with other histological hallmarks of NASH, such as balloon degeneration and sinusoidal fibrosis. Third, there should be evidence of progressive liver injury in association with continued insult. Currently, no animal model fulfills all three criteria.

Previous efforts to develop animal models of NAFLD have taken one of three main approaches: genetic models (e.g., *ob/ob* or *db/db*), dietary administrations (HFHC, WTDF, HFMCD), or a combination of the two. Most of these approaches have used rodents as the model animal.

An animal model using high-fat high-cholesterol diet-fed rabbits, which leads to hepatic cholesterol accumulation and progressive liver fibrosis and early cirrhosis. This approach may serve as a useful tool to study the role of free and total cholesterol in progressive liver injury and the molecular basis of fibrogenesis in fatty liver disease. The use of different animal models can make up for each other's shortcomings so as to achieve the goal of simulating the pathological characteristics of human NAFLD as much as possible.

Minor concerns:

Question 1:

1. Multiple animal models were used in this study, and tissue staining showed that NASH was aggravated after TRIM26 knockdown. Authors should show the representative livers of disease to prove whether liver changed in appearance.

Response 1:

Thank you so much for your concerns here. The corresponding representative liver appearance pictures regarding *Trim26*-deletion-related experiments have been placed in revised Supplementary figure 4 E, 5 E, 7 D.

Supplementary figure 4 E

Supplementary figure 5 E

Supplementary figure 7 D

Question 2:

2. WB results in this research can show the expression trend correctly, but almost all WB results have the phenomenon of tailing, which may be caused by high temperature when transferring or excessive residual fat in the process of protein extraction. It is need to extract the protein for WB again, so as to facilitate the display of the results more clearly and effectively.

Response 2:

Thank you very much for comment here. According to your suggestion, we have re-extracted the protein to re-performed western blotting analysis for key indicators in this study. Also, all the raw WB bands have been submitted along with our revised files.

Question 3:

3. All proteins should be capitalized, such as TRIM26, CEBPD (Fig.6A,6B, Fig.7C,7E, etc.)

Response 3:

This issue has been corrected in accordance with your comments.

Question 4:

4. Appropriate magnification should be applied in the immunofluorescent staining according to Fig. 6I

Response 4:

This point has been revised in accordance with your comments.

Question 5:

5. In Fig. S3, 'AdTrim26-treasfected' was wrong written, it should be AdTrim26- transfected.

Response 5:

We are very sorry for the incorrect marks and these misleading points have been corrected.

Question 6:

6. English should be improved by native speaking professional editor.

Response 6:

Thank you very much for the reviewer's comments here. We have revised the whole manuscript to correct the corresponding issues you mentioned above. The revised sections or sentences have been marked in our revised file for your consideration. We hope these changes meet your requirements.

Reviewers' Comments:

Reviewer #1:

Remarks to the Author:

Thank you for the responses provided. The authors addressed some of the items in their responses but clarity should be provided in the manuscript as well.

Specifically, with regards to comment/question 1 - the process for exclusion of autoimmune hepatitis -- if not done, should be explicitly stated given the proposed mechanism of TRIM26. Also the process for excluding other causes of steatosis (PBC, rare metabolic disease, etc) should be named as performed or not performed. The authors comment of 'tried our best' needs to be explicitly enumerated as done or not done to reflect addressing known causes of hepatic steatosis. These changes were not made in the materials and methods.

With regards to comment/question 2, please identify changes within the manuscript to reflect these additional clarifying pertinent negatives (ie other lipid lowering, glucose modulating, anti-inflammatory pharmacologic therapies were not excluded)

Noting why these exclusions are relevant for the discussion (interaction with TRIM26) is also needed as may limit impact of findings.

Reviewer #2:

Remarks to the Author:

Dear Authors,

Thank you for addressing my question about the NASH scoring.

Congratulations with your manuscript.

I will recommend it for publication.

Your sincerely,

Reviewer #3:

Remarks to the Author:

I truly thank the Authors for their efforts in revising the manuscript according to reviewers' suggestions.

I think that they have already responded to all my queries, therefore in my opinion the revised version is acceptable for the publication in this prestigious journal.

Best regards

Reviewer #4:

Remarks to the Author:

Thanks for the authors careful responses to the questions, they indeed answered all questions about the article with sufficient evidence, and I suggest the paper be published.

Response to the reviewers comments

Reviewer #1 (Remarks to the Author):

Thank you for the responses provided. The authors addressed some of the items in their responses but clarity should be provided in the manuscript as well.

Specifically, with regards to comment/question 1-the process for exclusion of autoimmune hepatitis -- if not done, should be explicitly stated given the proposed mechanism of TRIM26. Also the process for excluding other causes of steatosis (PBC, rare metabolic disease, etc) should be named as performed or not performed. The authors comment of 'tried our best' needs to be explicitly enumerated as done or not done to reflect addressing known causes of hepatic steatosis. These changes were not made in the materials and methods.

With regards to comment/question 2, please identify changes within the manuscript to reflect these additional clarifying pertinent negatives (i.e., other lipid lowering, glucose modulating, anti-inflammatory pharmacologic therapies were not excluded)

Noting why these exclusions are relevant for the discussion (interaction with TRIM26) is also needed as may limit impact of findings.

Response:

Thank you so much for your additional comments here. According to your concerns, we listed more detailed exclusion criteria in Method section-Human Donors Liver Samples to ensure the accuracy of studies on human samples (indicated below). Also, the possible influencing factors regarding lipid lowering, glucose modulating, anti-inflammatory pharmacologic therapies were also added to the exclusion criteria.

Statement in manuscript:

steatotic liver samples from patients with any of the following conditions were excluded from the study: (1) excessive drinking (alcohol >70 g for female or alcohol >140 g for male, per week), (2) viral infection, drug abuse (including hepatitis B & C virus infection), lipid-lowering, blood glucose regulation, anti-inflammatory drugs treatment (≥ 24 months) (3) autoimmune hepatitis, and (4) other causes of steatosis (*e.g.*, primary biliary cirrhosis, rare metabolic disease, etc).

Finally, thank you so much for your efforts to review and improve our current study.

Reviewer #2 (Remarks to the Author):

Dear Authors,

Thank you for addressing my question about the NASH scoring.

Congratulations with your manuscript.

I will recommend it for publication.

Your sincerely,

Thank you so much for your efforts to review and improve our current study.

Reviewer #3 (Remarks to the Author):

I truly thank the Authors for their efforts in revising the manuscript according to reviewers' suggestions.

I think that they have already responded to all my queries, therefore in my opinion the revised version is acceptable for the publication in this prestigious journal.

Best regards

Thank you so much for your efforts to review and improve our current study.

Reviewer #4 (Remarks to the Author):

Thanks for the authors careful responses to the questions, they indeed answered all questions about the article with sufficient evidence, and I suggest the paper be published.

Thank you so much for your efforts to review and improve our current study.